# Specific cancer-associated mutations in the switch III region of Ras increase tumorigenicity by nanocluster augmentation

**Maja Šolman[1], Alessio Ligabue[1], Olga Blaževitš[1], Alok Jaiswal[2], Yong Zhou[3], Hong Liang[3], Benoit Lectez[1], Kari Kopra[4], Camilo Guzmán[1], Harri Härmä[4], John F Hancock[3], Tero Aittokallio[2], Daniel Abankwa[1]***

[1]Turku Centre for Biotechnology, Åbo Akademi University, Turku, Finland; [2]Institute for Molecular Medicine Finland, University of Helsinki, Helsinki, Finland; [3]Department of Integrative Biology and Pharmacology, University of Texas Health Science Center at Houston, Houston, United States; [4]Department of Cell Biology and Anatomy, Institute of Biomedicine, University of Turku, Turku, Finland

**Abstract** Hotspot mutations of Ras drive cell transformation and tumorigenesis. Less frequent mutations in Ras are poorly characterized for their oncogenic potential. Yet insight into their mechanism of action may point to novel opportunities to target Ras. Here, we show that several cancer-associated mutations in the switch III region moderately increase Ras activity in all isoforms. Mutants are biochemically inconspicuous, while their clustering into nanoscale signaling complexes on the plasma membrane, termed nanocluster, is augmented. Nanoclustering dictates downstream effector recruitment, MAPK-activity, and tumorigenic cell proliferation. Our results describe an unprecedented mechanism of signaling protein activation in cancer.

*For correspondence:
daniel.abankwa@btk.fi

**Competing interests:** The authors declare that no competing interests exist.

## Introduction

The small GTPase Ras is dynamically anchored to cellular membranes. Its principal steady-state localization is the plasma membrane, from where most of Ras signaling emerges (*Hancock., 2003*). Ras hyperactivation is a hallmark of cancer and has long been considered to be pharmacologically intractable (*Cox et al., 2014*). However, recent progress encourages the pursuit of specific Ras protein inhibitors (*Spiegel et al., 2014*).

Since the seminal publication of Scheffzeck et al., it is well understood that hot-spot mutations in codons 12, 13, and 61 (*Scheffzek et al., 1997*), which account for >99% of Ras mutations (*Prior et al., 2012*), primarily compromise its GAP-dependent hydrolase activity. These oncogenic lesions render Ras transforming and tumorigenic, as they leave Ras constitutively GTP bound and therefore ready to overdrive in particular the Ras/MAPK pathway, which critically controls cell division and other oncogenic processes. A different mutational spectrum is found in RASopathies, developmental syndromes that are characterized by facio-cutaneous malformations, heart developmental and neurodevelopmental defects, as well as a predisposition to certain cancers (*Schubbert et al., 2007*). These syndromes are also characterized by a hyperactivation of the Ras/MAPK-signaling pathway. For example, in Noonan syndrome, *NRAS* or *KRAS* can be mutated at various positions along their coding sequences in the germline. The exact molecular and cellular mechanisms that lead to the observed phenotypes are still largely unclear (*Prior et al., 2012*). For non hot-spot mutations in Ras that coincide with the known nucleotide binding regions, the G1–G5 boxes, mechanistic explanations for

**eLife digest** Cancer is a disease that develops when cells within the body acquire genetic mutations that allow them to grow and divide rapidly. Many human cancers have mutations in a gene that encodes a protein called Ras, which promotes cell growth and division by controlling the activities of other proteins.

Ras congregates at the membrane that surrounds the cell and can assemble into clusters (called nanoclusters) that each contain six to eight Ras proteins. The tight packing of the proteins in these nanoclusters increases the amount of Ras in the membrane locally, which allows Ras to interact with other proteins more efficiently to promote growth and cell division.

In normal cells, other proteins control when Ras is active. However, in many cancer cells, Ras is active all the time due to mutations that occur in three 'hotspots' within its gene. Other mutations in the gene that encodes Ras are also found in cancer cells, but these are less common and it is not clear how they alter the activity of the protein.

Here, Solman et al. used microscopy and biochemical techniques to study the effects of some of the less common mutations on Ras activity in human cells. The experiments show that several mutations that alter a region of Ras called the 'switch III region' moderately increase the activity of Ras. The mutations probably alter the way that Ras sits in the membrane, which in turn changes the way it interacts with other proteins and the membrane so that more Ras nanoclusters form.

Solman et al.'s findings reveal a new way that Ras can be activated in cancer cells. The next challenge is to develop drugs that block the formation of Ras nanoclusters and to find out if they have the potential to be used to treat cancer.

aberrant activities have been demonstrated or proposed (*Schubbert et al., 2007*; *Gremer et al., 2011*; *Prior et al., 2012*; *Cirstea et al., 2013*). Whether and how additional mutations across the remainder of the coding sequence of Ras affect its pathogenic activity is largely unknown.

Ras activity emerges in the plasma membrane, where 20–50% of Ras proteins are organized into isoform-specific, dynamic proteo-lipid complexes that contain 6–8 Ras proteins, termed nanocluster (*Abankwa et al., 2007*). The tight packing of this signaling protein increases its concentration locally and thus enables more efficient effector recruitment (*Rotblat et al., 2010*; *Guzmán et al., 2014b*). It was proposed that nanoclustering is a basic systems-level design principle for the generation of high-fidelity signal transduction (*Tian et al., 2007*). Essentially only three regulators (galectin-1 [Gal-1], galectin-3, and nucleophosmin) of Ras nanoclustering, so called nanocluster scaffolds, are known. The lectin Gal-1 is the best characterized nanocluster scaffold, which increases H-ras-GTP nanoclustering and effector recruitment, effectively by stabilizing immobile H-ras-GTP nanocluster (*Rotblat et al., 2010*).

We previously revealed another aspect of Ras membrane organization, showing that a novel switch III in Ras is somehow coupled to the reorientation of H-ras on the membrane (*Figure 1—figure supplement 1*). Mutations in the switch III and the structural elements of H-ras that stabilize its reorientation (helix α4 and the C-terminal hypervariable region [hvr]) systematically modulate Ras signaling (*Gorfe et al., 2007*; *Abankwa et al., 2008b, 2010*). More recently, we addressed the mechanistic basis of this activity modulation for computational modeling-derived mutations on helix α4 and the hvr: these alter engagement of the nanocluster modulator Gal-1 and thus H-ras nanoclustering. As a consequence of this up-concentration, effector recruitment and subsequent downstream signaling are increased (*Guzmán et al., 2014b*).

Here, we report that cancer-associated mutations in the switch III region of the three major Ras oncoproteins, H-, N-, and K-ras, increase Ras activity by a novel disease mechanism, namely signaling protein nanocluster augmentation. We find that these mutations do not alter basic biochemical functions of Ras in solution. Instead, a strict correlation between increased recruitment of the effector to Ras and augmented nanoclustering of Ras on cellular membranes is found. Upregulated effector engagement is directly reflected in the elevated cellular Ras activity, and significantly impacts on the tumorigenic potential. Our results reveal a new mechanism of mutational signaling pathway hyperactivation in a pathophysiological setting and suggest Ras nanoclusters as direct drug targets.

## Results

### The switch III region of H-ras couples to G-domain reorientation

H-ras exists in a nucleotide-dependent conformational equilibrium on the membrane (*Gorfe et al., 2007*; *Abankwa et al., 2008b*). The two delimiting conformers are stabilized by either helix α4 or the hvr (*Figure 1—figure supplement 1*). Conformer reorientation on the membrane was associated with a novel switch III region, which is formed by the β2-β3-loop and helix α5. However, formal proof for their mechanistic connection is still missing.

We previously found that mutations in the hvr or on helix α4 (left and right tables on top in *Figure 1A*) alter the activity of GTP-H-ras, probably by stabilizing preferred conformers (red and blue GTP-H-ras conformers, respectively, in *Figure 1A*) similar to the nucleotide-dependent ones (*Gorfe et al., 2007*; *Abankwa et al., 2008b*). More recent evidence from these helix α4 and hvr GTP-H-ras mutants suggests that the conformational state couples to nanoscale Ras-signaling hubs in the membrane, termed nanocluster (middle in *Figure 1A*) (*Guzmán et al., 2014b*). Nanoclustering then critically determines the recruitment rate of the effector Raf from the cytoplasm to membrane bound Ras and therefore the initiating event of the MAPK-signaling cascade (bottom in *Figure 1A*).

In order to validate that the switch III region is coupled intramolecularly to the reorientation of H-ras, we combined activating mutations in the switch III region with inactivating ones on helix α4 or inactivating switch III mutations with activating ones in the hvr (center table on top, *Figure 1A*). We predicted that in the case of intramolecular (allosteric) coupling these mutations would at least partially neutralize each other, as we have previously observed for mutant H-rasG12V-R128A,R135A, R169A,K170A (*Abankwa et al., 2008b*, *2010*), which combines the inactivating/neutralizing mutations R128A,R135A on helix α4 and activating ones of the hvr. Mutation R128A,R135A decreases H-rasG12V nanoclustering (*Guzmán et al., 2014b*), however, on its own it has only a small decreasing or neutral effect on H-ras activity (*Abankwa et al., 2008b*, *2010* and *Figure 1C*).

To measure the specific activity of H-ras mutants in mammalian cell lines, we used our well-established effector-recruitment FRET assay (*Abankwa et al., 2008b*, *2010*; *Guzmán et al., 2014b*) (*Figure 1B*). We coexpressed mGFP-tagged H-ras mutants and the mRFP-tagged Ras binding domain of C-Raf (in the following in short RBD) in BHK cells and monitored the change of FRET using fluorescence lifetime imaging microscopy (FLIM) (*Figure 1C,D*).

In agreement with our model of switch III coupling to the orientation of Ras on the membrane (*Abankwa et al., 2008a*, *2008b*), activating mutations D47A,E49A in the switch III region were neutralized by inactivating mutations R128A,R135A on helix α4, while the inactivating mutation R161A in switch III was neutralized by activating R169A,K170A in the hvr (*Figure 1C*). These results therefore agree with an allosteric coupling of the switch III region to the reorientation of H-ras on the membrane. Furthermore, they imply that mutations in the switch III could alter Ras nanoclustering, possibly by a conformation-associated mechanism, which we are however not going to focus on in the following, but discuss in detail at the end.

### Computational modeling-derived switch III mutations affect H-ras signaling output by increasing nanocluster

We next explored whether altered nanoclustering also underlies the activity changes that we previously observed for computational modeling-derived switch III mutant H-rasG12V-D47A,E49A (*Abankwa et al., 2008b*). We compared the nanoclustering of this hyperactive mutant to that of the parent H-rasG12V by statistical analysis of the distribution of immunogold labeled H-ras in electron microscopic images of plasma membrane sheets from BHK cells (*Plowman et al., 2005*). We observed a significant increase in nanoclustering of this modeling-derived mutant, suggesting that increased signaling originates from augmented nanoclustering (*Figure 2A*). In order to corroborate these data, we used our recently established Gal-1-dose dependent nanoclustering-response assay (*Guzmán et al., 2014b*). This assay measures the dependence of H-ras-mutant nanoclustering on different cellular levels of the nanocluster scaffold Gal-1. Due to the tight packing of mGFP- and mCherry-tagged Ras proteins in nanoclusters FRET emerges, thus serving as a read-out for nanoclustering (*Figure 2B*). As compared to H-rasG12V, nanoclustering-FRET of the modeling-derived switch III mutant H-rasG12V-D47A,E49A was significantly increased at all Gal-1 levels (*Figure 2C,D*); this was also reflected by the increased complexation of this mutant with Gal-1 in the cells (*Figure 2—figure supplement 1A*), similar to what

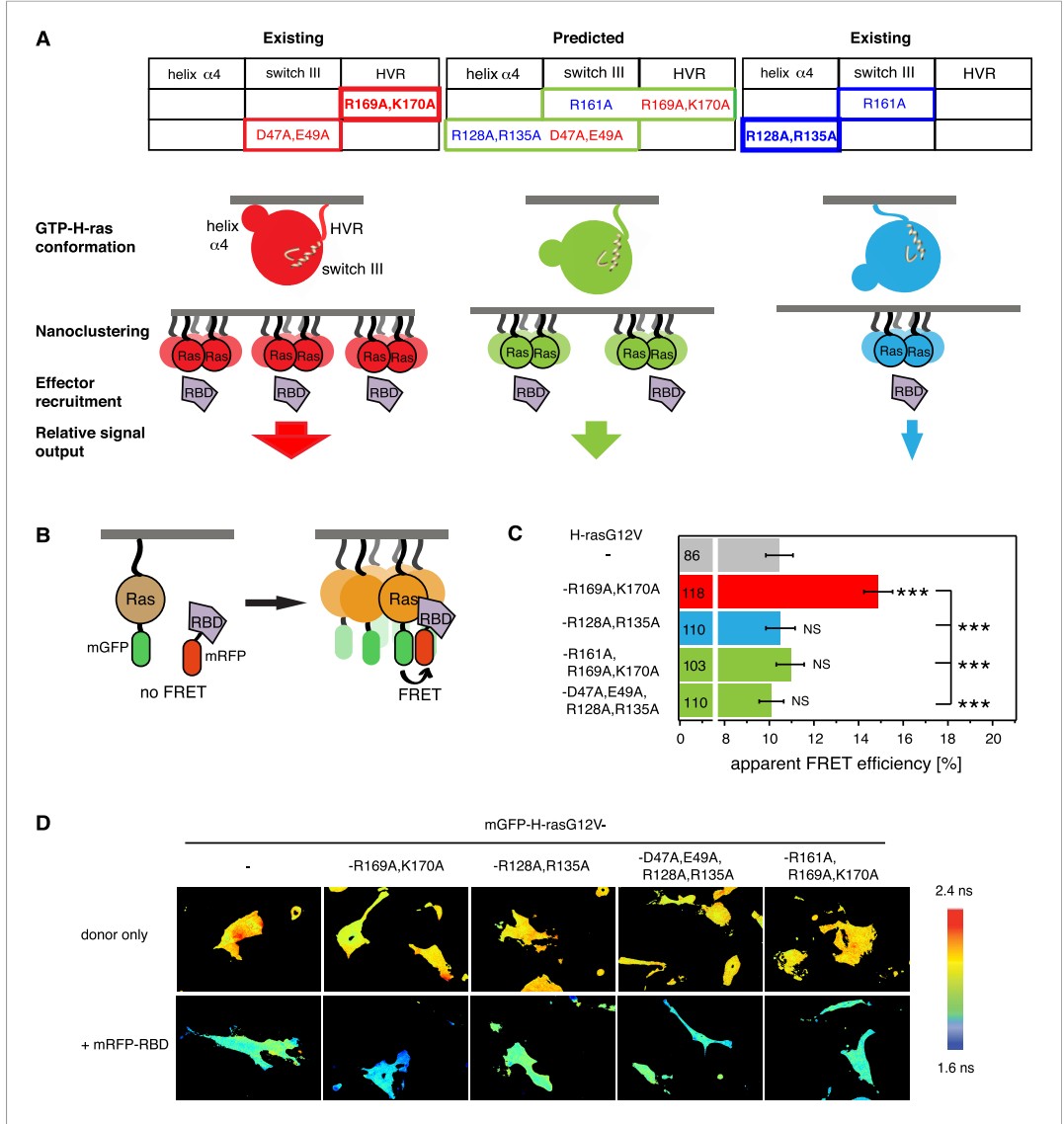

**Figure 1**. Intramolecular switch III-conformer coupling suggests altered nanoclustering in switch III mutants. (**A**) Schematic representation of conformer-nanocluster coupling and its effect on H-ras signaling. For GTP-Ras a conformational equilibrium is assumed, which can be shifted by mutations as indicated by matching colors (top tables, show mutations by region in Ras). Mutations in the table increase (red) or decrease/neutralize (blue) H-ras activity. For mutations boxed in bold this was previously shown to proceed by increasing or decreasing H-ras nanoclustering, respectively. As a consequence, the effector recruitment and thus downstream signaling are correspondingly modulated. Note, that a direct effect of the Ras conformation on effector engagement is not excluded. The green scenario represents the normal activity, which is predicted for the combinations of activity increasing and decreasing/neutralizing mutations (green scenario in centre), if switch III couples to H-ras reorientation. (**B**) Schematic representation of the RBD-recruitment FRET assay. Ras activity was measured by mRFP-C-Raf-RBD recruitment to mGFP-tagged mutants of Ras in intact cells using FRET. (**C**) RBD-recruitment FRET data of indicated H-ras mutants transiently expressed in BHK cells. Numbers in bars give number of analyzed cells from three independent experiments. Error bars represent the standard error of the mean (±SEM). The color code of the bars is as in (**A**) and indicates if the mutations increased, decreased/neutralized or normalized RBD-recruitment as compared to H-rasG12V. Statistical analysis of differences vs H-rasG12V was performed as described in 'Materials and methods' (NS, non-significant; ***, p < 0.001). (**D**) FRET in (**C**) was measured by fluorescence lifetime microscopy of transfected BHK cells as indicated. Color look up table to the right shows fluorescence lifetimes. See also *Figure 1—figure supplement 1*.

The following figure supplement is available for figure 1:

**Figure supplement 1**. Nucleotide-dependent H-ras conformers on the membrane.

we previously observed with a hvr-mutant (*Abankwa et al., 2010*). Taken together, these data confirmed that nanoclustering is increased in H-rasG12V-D47A,E49A.

We next examined, whether the nanoclustering response is also reflected by the recruitment of the effector C-Raf, as predicted by our recent computational simulations of nanoclustering (*Guzmán et al., 2014b*). In line with this simulation, RBD-recruitment to the structure modeling-derived switch III H-ras mutant was significantly increased at all Gal-1 levels, as compared to the parent H-rasG12V (*Figure 2E*). Correspondingly, Raf, MEK, and ERK phosphorylation was significantly increased when H-rasG12V-D47A,E49A was expressed, suggesting that increased nanoclustering was propagated all the way through the MAPK-signaling pathway (*Figure 2F*; *Figure 2—figure supplement 1B*).

Importantly, enhanced effector-recruitment (*Figure 2E*), which initiates the signal cascade was not due to an overall increase of the affinity to the Ras mutant, as revealed in anisotropy binding experiments of purified GTPγS-H-ras-D47A,E49A to the RBD in vitro (*Figure 2G*, *Supplementary file 1*). Also GAP-mediated hydrolysis (*Figure 2H*, *Figure 2—figure supplement 1C*) was effectively unaltered as compared to wt H-ras. A relatively small decrease in the SOS-mediated GTP-analog off-rate and consistently an increase in EGF-stimulated activation however suggested that this mutant may be more efficiently activated (*Figure 2—figure supplement 2*, *Supplementary file 1*).

We conclude that increased effector recruitment (*Figure 2E*) and signaling (*Figure 2F*) of H-rasG12V with the structure modeling-derived mutation D47A,E49A in the switch III region is due to augmented Ras nanoclustering.

## Analysis of orientation-switch III mutations using cancer genomic databases

The computational modeling-derived switch III residues D47 and E49 are phylogenetically highly conserved (*Figure 3—figure supplement 1*), consistent with a biologically significant role of their structural context. We hypothesized that if our link between the switch III and nanoclustering was physiologically relevant, genetic diseases with hyperactivated Ras, such as many cancers, should have exploited it to upregulate Ras-signaling activity.

We therefore analyzed more than 140,000 cancer sample entries from public cancer genomic databases (COSMIC, cBioPortal and ICGC) for point mutations in the orientation-switch III region of *HRAS*, *NRAS*, and *KRAS* genes; specifically, in the β2-β3-loop and helix α5 (switch III), as well as the orientation stabilizing elements, the helix α4 and the hvr. We found that approximately 12% of the distinct mutations along the Ras sequences occur in these switch III regions. However, the overall incidence is very low with a total number of 15, 20, and 28 reported cases for H-, N-, and K-ras, respectively (*Supplementary file 2*).

Despite the low number of cases, we analyzed the isoform-specific mutations by region and by cancer type (*Figure 3*, *Supplementary file 2*). For H-ras colorectal as well as hematopoietic and lymphoid tissue samples showed the highest number of cases with orientation-switch III mutations (3 each), while for N-ras, we found colorectal and skin tumor samples with a slightly higher occurrence of 5 cases each reported. In agreement with the overall highest mutation frequency of K-ras in cancer (*Prior et al., 2012*), the highest total number of mutations was found in this Ras isoform, with the highest incidence numbers in colorectal and endometrial samples. In total, it appeared that orientation-switch III mutations occur most frequently in colorectal cancer samples.

When analyzed by affected region, it became apparent that certain parts of the orientation-switch III are more frequently hit in a given isoform. In H-ras helix α4, in N-ras the β2-β3-loop, and in K-ras the helix α5 are most frequently affected by point mutations (*Figure 3*, *Supplementary file 2*). To the best of our knowledge, it is currently unknown whether and by which mechanism these mutations affect tumorigenesis.

## A skin tumor associated mutation in the switch III region of H-ras increases effector recruitment by augmented nanoclustering

Based on our mechanistic insight, we hypothesized that these cancer-associated orientation-switch III mutations increase Ras signaling by augmenting its nanoclustering. Only nanoclustering of GTP-loaded Ras is relevant for its signaling output (*Tian et al., 2007*). We therefore clamped all mutations in the GTP-state, by co-introduction of the G12V mutation. This also allowed us to compare

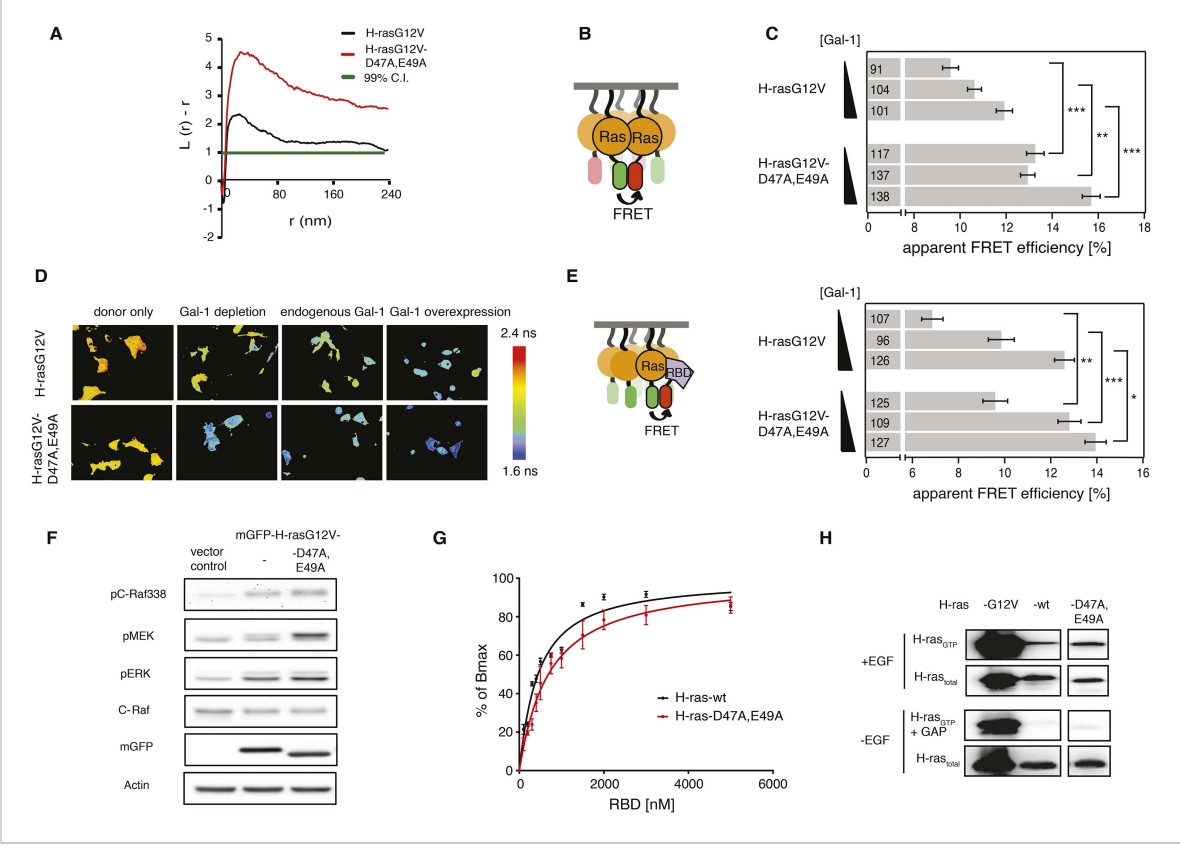

**Figure 2**. Computational modeling-derived switch III mutations D47A,E49A in H-ras increase nanoclustering and RBD-recruitment. (**A**) Electron microscopic nanoclustering analysis of mGFP-H-rasG12V and mGFP-H-rasG12V-D47A,E49A in BHK cells. Normalized univariate K-functions, where maximal $L(r)$-$r$ values above the 99% CI for complete spatial randomness indicate clustering at that value of r (number of membrane sheets analyzed per condition, n = 17). (**B**) Schematic representation of nanoclustering-FRET analysis, where mGFP-tagged and mCherry-tagged Ras constructs were co-expressed in cells. (**C**) The nanoclustering-FRET response of H-rasG12V-D47A,E49A and its parent construct in dependence of the dose of the nanocluster scaffold Gal-1 in BHK cells. (**D**) Representative nanoclustering-FRET fluorescence lifetime images of BHK cells expressing FRET-pairs (or donor only, left column) of constructs indicated on the left, under Gal-1 conditions as annotated on the top. Color look up table to the right shows fluorescence lifetimes. (**E**) RBD-recruitment FRET data of H-rasG12V-D47A,E49A and its parent construct at three different Gal-1 doses analyzed using FRET-imaging of transiently transfected BHK cells. (**C**, **E**) Numbers in bars give numbers of analyzed cells from three independent experiments. Error bars represent the standard error of the mean (±SEM). Statistical analysis vs parent RasG12V was performed as described in 'Materials and methods' (*p < 0.05; **p < 0.01; ***p < 0.001). (**F**) Western blot analysis of C-Raf, MEK, and ERK phosphorylation and C-Raf in BHK cells transiently expressing mGFP-H-rasG12V-D47A, E49K, its parent or an empty vector control. Equal expression of Ras constructs can be seen in the mGFP row, equal loading in the Actin row. (**G**) In vitro RBD binding to mant-GTPγS loaded wild-type (wt) H-ras or H-ras-D47A,E49A measured with a fluorescence anisotropy assay. Details on the fitting function are in the 'Materials and methods'. Data are averages ±SEM of three repeats. (**H**) RBD-pulldown experiments in BHK cells transiently expressing indicated H-ras mutants or wt H-ras. Top panel shows level of active Ras after EGF-stimulation (100 ng/ml). Bottom panel shows GAP sensitivity of wt and mutant H-ras proteins. See also *Figure 2—figure supplement 1* and *Figure 2—figure supplement 2*.

The following figure supplements are available for figure 2:

**Figure supplement 1**. The computational modeling-derived switch III H-ras mutant exhibits stronger Gal-1 complexation and remains sensitive to GAP-mediated hydrolysis.

**Figure supplement 2**. SOS-mediated nucleotide exchange kinetics, as well as mantGTPγS dissociation constants of wt H-ras and mutants.

results with those of the computational modeling-derived mutants (*Figures 1, 2*). We then ensured by a comprehensive set of biochemical experiments that the same mutations in the wild type background did not alter essential activities of Ras, such as GTP-loading, GTP-hydrolysis or effector binding, which classically increase Ras activity.

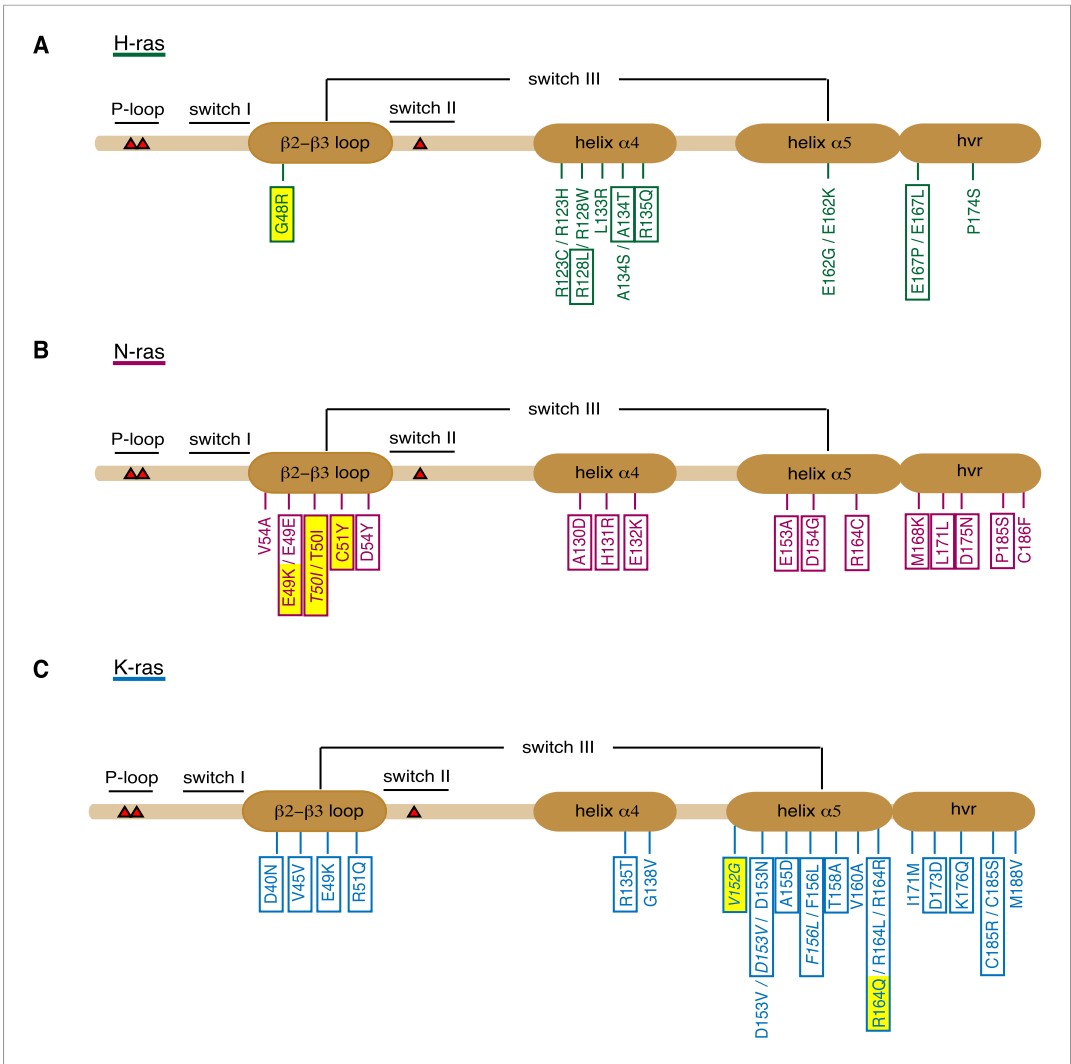

**Figure 3**. Orientation-switch III mutations in oncogenic Ras isoforms that are reported in cancer genome databases. Cancer-associated point mutations (missense, nonsense, and silent) in the orientation-switch III regions as reported in COSMIC, cBioPortal, and ICGC databases. Schematic representations of the linear protein structures of (**A**) H-ras, (**B**) N-ras, and (**C**) K-ras. Critical functional regions in the structures are annotated; those of the orientation-switch III by ovals. Approximate position of hot-spot mutation residues (G12, G13, and Q61) is marked with red triangles. Mutations from cancer patient samples are boxed, RASopathy mutations are boxed and in italics, other annotated mutations are from cancer cell lines (*Prior et al., 2012*). Mutations that are studied here are highlighted in yellow. See also *Figure 3—figure supplement 1*.

The following figure supplement is available for figure 3:

**Figure supplement 1**. Phylogenetic analysis of the Ras switch III region.

Given that the switch III region is most frequently mutated across all Ras isoforms, we focused on patient samples with mutations in this region (*Figure 3*). In H-ras only one case of a switch III region mutation is described in an atypical spitzoid tumor (AST) sample. Curiously, this mutation G48R is found together with a second D92N-mutation in H-ras.

Confocal colocalization analysis confirmed that the localization of both mutants in BHK cells was unaltered as compared to parent H-rasG12V (*Figure 4A*). Assuming that the G48R,D92N mutations were gain of function, we combined these cancer-associated mutations with the inactivating mutation R128A, R135A on helix α4. This new mutant showed normalized H-ras activity (*Figure 4—figure supplement 1A*),

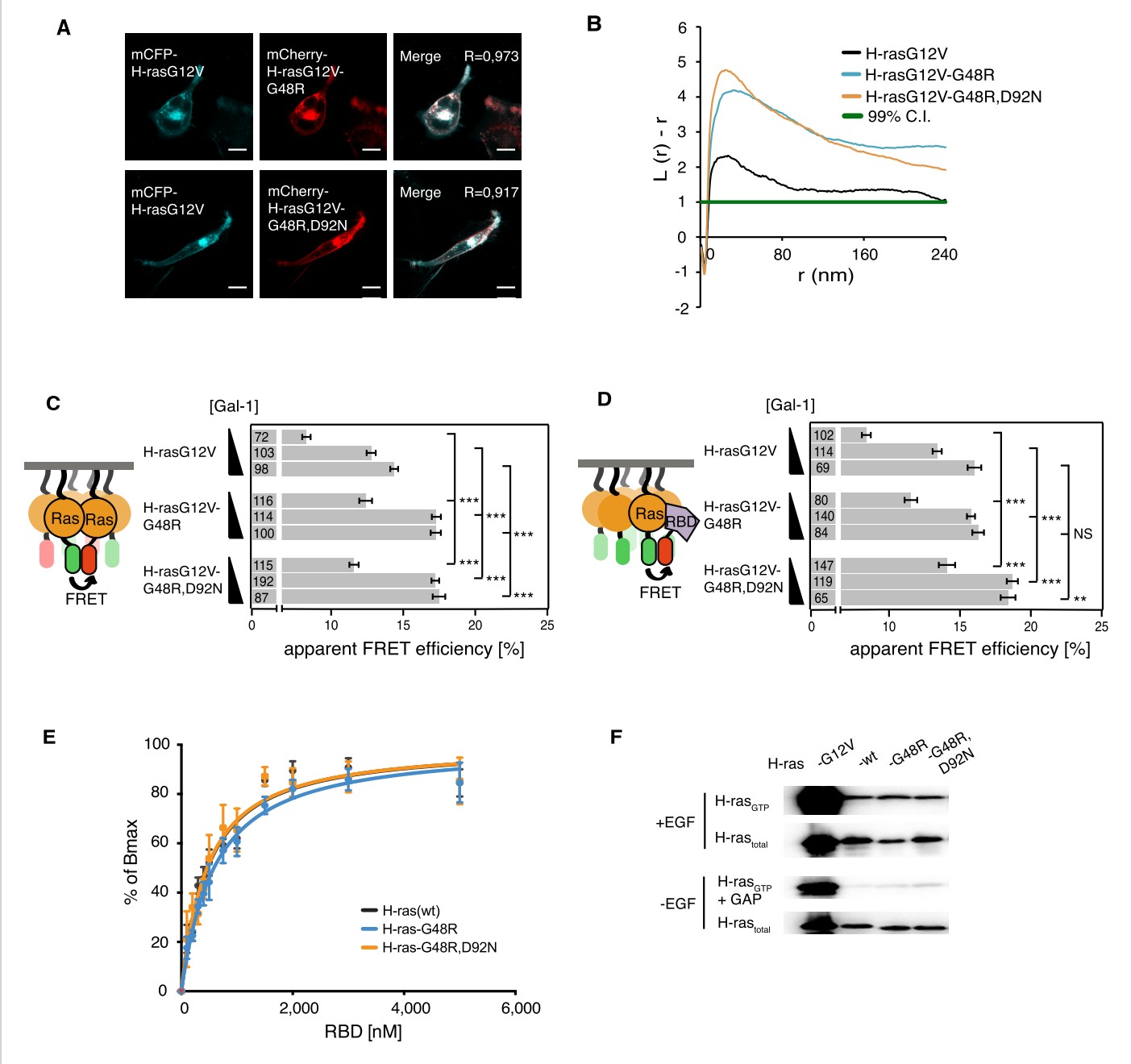

**Figure 4**. Increased nanoclustering and effector recruitment by a tumor-derived switch III mutation in H-ras. (**A**) Representative confocal images of mCFP-H-rasG12V co-localization with the switch III mutants mCherry-H-rasG12V-G48R (top) or mCherry-H-rasG12V-G48R,D92N (bottom) in BHK cells. The colocalization Manders' overlap coefficient (*R*) is marked on the merged images. Scale bar is 10 μm. (**B**) Electron microscopic nanoclustering analysis of mGFP-tagged H-ras mutants in BHK cells. Normalized univariate K-functions, where maximal *L(r)-r* values above the 99% CI for complete spatial randomness indicate clustering at that value of r (number of membrane sheets analyzed per condition, n > 17). (**C**) The nanoclustering-FRET response of the tumor-derived H-rasG12V-G48R,D92N mutant, its switch III mutation-only derivative H-rasG12V-G48R and its parent construct in dependence of the dose of the nanocluster scaffold Gal-1 in BHK cells. (**D**) RBD-recruitment FRET analysis of H-rasG12V with or without mutations G48R or G48R,D92N at three different Gal-1 doses in BHK cells. (**C**, **D**) Numbers in bars give number of analyzed cells from three independent experiments. Error bars represent the standard error of the mean (±SEM). Statistical analysis vs parent RasG12V was performed as described in the 'Materials and methods' (NS, non-significant; **p < 0.01; ***p < 0.001). (**E**) In vitro RBD binding to mant-GTPγS loaded wild-type (wt) H-ras or H-ras carrying mutations G48R or G48R,D92N measured with a fluorescence anisotropy assay. Binding constants are given in *Supplementary file 1*. Note the orange curve covers the black. Details on the fitting function are in the 'Materials and methods'. Data are averages ±SEM of three repeats. (**F**) RBD-pulldown experiments in BHK cells transiently expressing indicated H-ras mutants or wild type (wt) H-ras. Top panel shows level of active Ras after EGF-stimulation (100 ng/ml). Bottom panel shows

*Figure 4. continued on next page*

*Figure 4. Continued*

GAP-sensitivity of wt and mutant H-ras proteins. Note that control lanes marked G12V and wt are identical to those in *Figure 2H*, as these H-ras samples were always run on the same gel. See also *Figure 4—figure supplement 1*.

The following figure supplement is available for figure 4:

**Figure supplement 1**. Cancer-associated H-ras switch III mutations have no relevant effect on its biochemical properties.

comparable to the modeling-derived orientation-switch III mutants (*Figure 1D*). This suggested that the cancer-associated mutation is also coupled to the reorientation mechanism of membrane bound H-ras and may thus also impact on its nanoclustering. Indeed, electron microscopic analysis of plasma membrane sheets derived from BHK cells expressing the single or double-mutant revealed a strong and significant increase in nanoclustering of both mutants as compared to the parent H-rasG12V (*Figure 4B*). This was confirmed by nanoclustering-FRET analysis in cells, where consistent with the increased Gal-1 complexation of the G48R-mutant (*Figure 4—figure supplement 1B*), the Gal-1-dose-dependent nanoclustering response was significantly increased for both the G48R and G48R,D92N mutants at all Gal-1 levels (*Figure 4C*). Moreover, the Gal-1-dose-dependent RBD recruitment response of both mutants was increased in cells (*Figure 4D*), while RBD-binding in vitro remained indistinguishable from the parent Ras (*Figure 4E*, *Supplementary file 1*). The correlation of the RBD- and nanoclustering-response pattern again suggested that augmented nanoclustering increased H-ras activity by enhancing effector recruitment. Together with the absence of any RBD-binding defect, this led us to the conclusion that introduction of mutations G48R or G48R,D92N enhances effector recruitment due to increased nanoclustering.

As observed for the modeling-derived switch III mutant, basic biochemical parameters of H-ras-G48R or H-ras-G48R,D92N, such as GAP-mediated GTP hydrolysis (*Figure 4F*, *Figure 4—figure supplement 1C*), as well as GTP affinity (*Supplementary file 1*) remained unaltered. However, mutations G48R,D92N, but not G48R alone, showed a ~1.4-fold elevation of SOS-mediated nucleotide exchange (*Supplementary file 1*) consistent with an increased activation upon EGF-stimulation (*Figure 4F*, *Figure 4—figure supplement 1C*). Therefore, in this cancer-derived H-ras mutation a combination of weak biochemical alterations and a significantly increased nanoclustering, which both result in an activating phenotype of the G48R,D92N mutation in H-ras, may explain the occurrence of these two H-ras mutations in the tumor.

## Two tumor-associated mutations in the switch III region of N-ras promote its transforming activity by increasing its nanoclustering

For unknown reasons, both N- and K-ras are more frequently mutated and associated with more aggressive cancers than H-ras. N-ras is highly associated with skin and hematological cancers, while K-ras is highly mutated in pancreatic and intestinal cancers (*Prior et al., 2012*). We therefore in the following concentrated on these isoforms to investigate the direct involvement of switch III mutations in tumor growth. In N-ras, the T50I mutation was found in a malignant melanoma and was recently also described to be associated with Noonan syndrome (*Cirstea et al., 2010*), a RASopathy. The additional two mutations, C51Y and E49K, were found in hematologic and skin malignancies, respectively. Interestingly, the highly conserved E49 is also mutated in K-ras in colorectal cancer (*Supplementary file 2*). Moreover, our data from the computational modeling-derived H-ras switch III mutant D47A, E49A already suggested that this residue might be involved in nanocluster augmentation.

As observed for H-ras switch III mutations, none of the three N-ras mutations altered the subcellular distribution of N-ras (*Figure 5A*). We next assessed nanoclustering and effector recruitment of these mutants in BHK cells. Gal-1 is not known to impact on N-ras nanoclustering and binding to it remained unaltered by the mutations in N-ras (*Figure 5—figure supplement 1A*). We therefore investigated the N-ras mutants only at endogenous Gal-1 concentrations. While mutation T50I had no significant effect, both mutation E49K and C51Y significantly increased nanoclustering (*Figure 5B,C*) and ensuing Ras effector recruitment as compared to parent N-rasG12V (*Figure 5D*).

In order to assess RBD-binding to cytoplasmic Ras, that is, in solution, we treated mammalian cells expressing Ras and RBD constructs with compactin, as compactin blocks prenylation and therefore

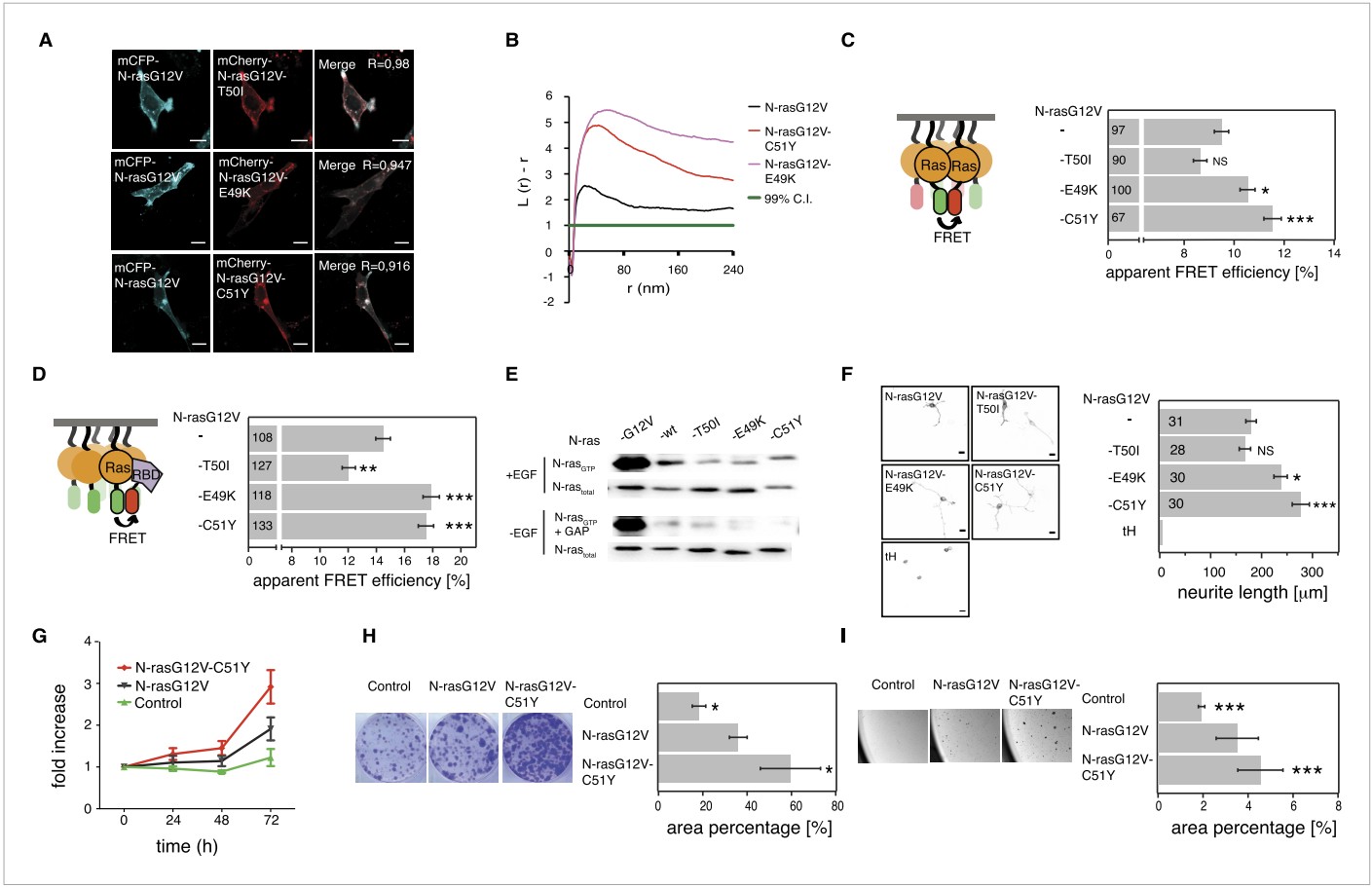

**Figure 5**. Tumor-derived N-ras switch III mutations increase cell proliferation and transforming activity by increased nanoclustering. (**A**) Representative confocal images of mCFP-N-rasG12V co-localization with the switch III mutants mCherry-N-rasG12V-T50I, mCherry-N-rasG12V-E49K, or mCherry-N-rasG12V-C51Y in BHK cells. The Manders' coefficient (*R*) that quantifies co-localization is marked on the merged images. Scale bar is 10 μm. (**B**) Electron microscopic nanoclustering analysis of mGFP-tagged N-ras mutants in BHK cells. Normalized univariate K-functions, where maximal *L(r)-r* values above the 99% CI for complete spatial randomness indicate clustering at that value of r (number of membrane sheets analyzed per condition, n > 17).
(**C**) Nanoclustering-FRET analysis of cancer-associated N-rasG12V-T50I, N-rasG12V-E49K and N-rasG12V-C51Y as compared to their parent construct in BHK cells. (**D**) RBD-recruitment FRET analysis of N-rasG12V-T50I, N-rasG12V-E49K, and N-rasG12V-C51Y as compared to their parent construct in BHK cells. (**E**) RBD-pulldown experiments in BHK cells transiently expressing indicated N-ras mutants or wild-type (wt) N-ras. Top panel shows level of active N-ras after EGF-stimulation (100 ng/ml). Bottom panel shows GAP-sensitivity of wt N-ras and mutant N-ras proteins. (**F**) MAPK-dependent PC12-differentiation assay. PC12 cells transiently expressed indicated mGFP-tagged N-rasG12V with or without switch III mutations, or the control tH (minimal membrane anchor of H-ras) for 48 hr. Subsequently neurite length was assessed (right). Examples of confocal images of PC-12 cells expressing indicated constructs used for neurite length quantification (left). Scale bar is 15 μm. (**C**, **D**, **F**) Numbers in bars give number of analyzed cells from three independent experiments. Error bars represent the standard error of the mean (±SEM). Statistical analysis vs parent RasG12V was performed as described in the 'Materials and methods' (NS, non-significant; *p < 0.05; **p < 0.01; ***p < 0.001). (**G**) Proliferation of NIH/3T3 cells stably expressing N-rasG12V, N-rasG12V-C51Y, or a control construct. Control is transduced with the GFP reporter only. Error bars represent the standard error of the mean (±SEM). (**H**) Transformation of NIH/3T3 cells stably expressing N-rasG12V as compared to N-rasG12V-C51Y or a vector-only control. Foci formed after 7 days of growth were stained with crystal violet and mean (±SEM) foci areas from three biological repeats were quantified (right). Representative images of crystal violet-stained foci of transduced NIH/3T3 cells (left). (**I**) Anchorage-independent growth of NIH/3T3 cells stably expressing N-rasG12V as compared to N-rasG12V-C51Y or a vector-only control. Colonies were imaged after 14 days (left), and the colony area was determined (right). Three independent biological repeats were performed and each repeat was done in triplicate. Error bars represent the standard error of the mean (±SEM). See also
*Figure 5—figure supplement 1*.

The following figure supplement is available for figure 5:

**Figure supplement 1**. Cancer-associated mutations in switch III of N-ras do not alter its biochemical properties and Gal-1 complexation.

membrane anchorage of Ras. We previously showed that these compactin-treatment experiments provide the same information as binding experiments with purified proteins (*Guzmán et al., 2014b*). All mutants bound the RBD indistinguishably from N-ras without switch III mutations (*Figure 5—figure supplement 1B*). Moreover, both hyperactive mutants N-ras-E49K and N-ras-C51Y, as well as N-ras-T50I remained sensitive to GAP-mediated hydrolysis and did not display increased background activation of MEK, ERK, or AKT in the absence of serum (*Figure 5E*, *Figure 5—figure supplement 1C,D*). Thus, basic biochemical parameters were largely preserved, confirming that increased RBD-recruitment followed augmented nanoclustering of N-ras with switch III mutations E49K or C51Y.

It was previously observed that RASopathy-associated Ras mutants do not lead to a clear increase of the downstream pathway (*Gremer et al., 2011*; *Cirstea et al., 2013*). Similar to these observations, neither GTP-loading (*Figure 5E*, *Figure 5—figure supplement 1C*) nor phosphorylation of MEK, ERK, and AKT (*Figure 5—figure supplement 1D*) was clearly increased after EGF-stimulation. We therefore resorted to more complex read-outs of Ras activity. Differentiation of rat adrenal pheochromocytoma (PC12) cells is accompanied by neurite outgrowth, which serves as a well established and sensitive measure for Ras/MAPK activity (*Gorfe et al., 2007*). PC12 cells transfected with mGFP-tagged mutant N-rasG12V-E49K or N-rasG12V-C51Y showed significantly increased neurite length (*Figure 5F*). The most active mutant N-rasG12V-C51Y was then further analyzed for its effect on the tumorigenic growth potential of cells. NIH/3T3 cells stably expressing this mutant (*Figure 5—figure supplement 1E*) showed significantly increased cell proliferation (*Figure 5G*), focus formation (*Figure 5H*), and anchorage-independent growth in soft agar (*Figure 5I*), which typically correlates with the tumorigenic potential in vivo.

In summary, cancer-derived mutations E49K and C51Y do not affect relevant biochemical functions of N-ras, but instead increase its nanoclustering potential to enhance effector recruitment and subsequent Ras/MAPK-signaling output. Our most active N-ras mutant furthermore revealed that augmented nanoclustering ultimately increases cell proliferation and tumorigenic growth of transformed cells.

## Increased oncogenic activity of nanoclustering augmenting mutation R164Q on helix α5 of K-ras

The *KRAS* gene is the developmentally most significant Ras isoform. Its important developmental role is furthermore underscored by the fact that activating germline mutations that are associated with the K-ras4B isoform (henceforth K-ras) lead to relatively severe RASopathy phenotypes (*Carta et al., 2006*). In addition to our N-rasT50I mutant, we therefore added the Noonan syndrome-associated mutation V152G on helix α5 of the switch III of K-ras to our otherwise cancer mutation-focused analysis. These two mutations represented mechanistically unresolved switch III mutations in RASopathies.

Like for the N-ras Noonan syndrome mutant, subcellular distribution remained unchanged (*Figure 6A*) and no increase in nanoclustering (*Figure 6B,C*), effector recruitment (*Figure 6D*) or PC12 differentiation (*Figure 6F*) was found. Also the binding of the mutant to the RBD in solution was unchanged compared to the parent K-ras (*Figure 6—figure supplement 1B*), and its GAP sensitivity and downstream signaling activity under serum starvation were unaltered (*Figure 6E*, *Figure 6—figure supplement 1C,D*).

We then turned to investigate the most frequent switch III mutation, R164Q in K-ras, which was found in colorectal and endometrial cancers. A previous study by others showed that this helix α5 mutation alone (i.e., in the wt background) did not possess any detectable increase in its transforming potential, yet gene expression analysis suggested a mild activation phenotype (*Smith et al., 2010*).

Addition of R164Q did not alter the subcellular distribution of mGFP-tagged K-rasG12V (*Figure 6A*) in BHK cells. Like for the N-ras mutants, binding to Gal-1 was unaltered (*Figure 6—figure supplement 1A*), which again led us to study the activity only at endogenous Gal-1 levels. Nanoclustering analysis of K-rasG12V-R164Q by electron microscopic- (*Figure 6B*) and by nanoclustering-FRET-analysis (*Figure 6C*) in BHK cells both confirmed significantly increased nanoclustering as compared to the parent K-ras. In agreement with our nanocluster-activity model (*Figure 1A*), augmented nano-clustering was strictly followed by an increased recruitment of the effector fragment RBD in BHK cells (*Figure 6D*). Importantly, in solution the effector fragment again was bound equally to the mutant and to the parent K-ras (*Figure 6—figure supplement 1B*). Together with the normal GAP-sensitivity of K-ras-R164Q (*Figure 6E*, *Figure 6—figure supplement 1C*), these results support that the mutation neither affected the affinity to the RBD of C-Raf nor GAP-mediated Ras inactivation.

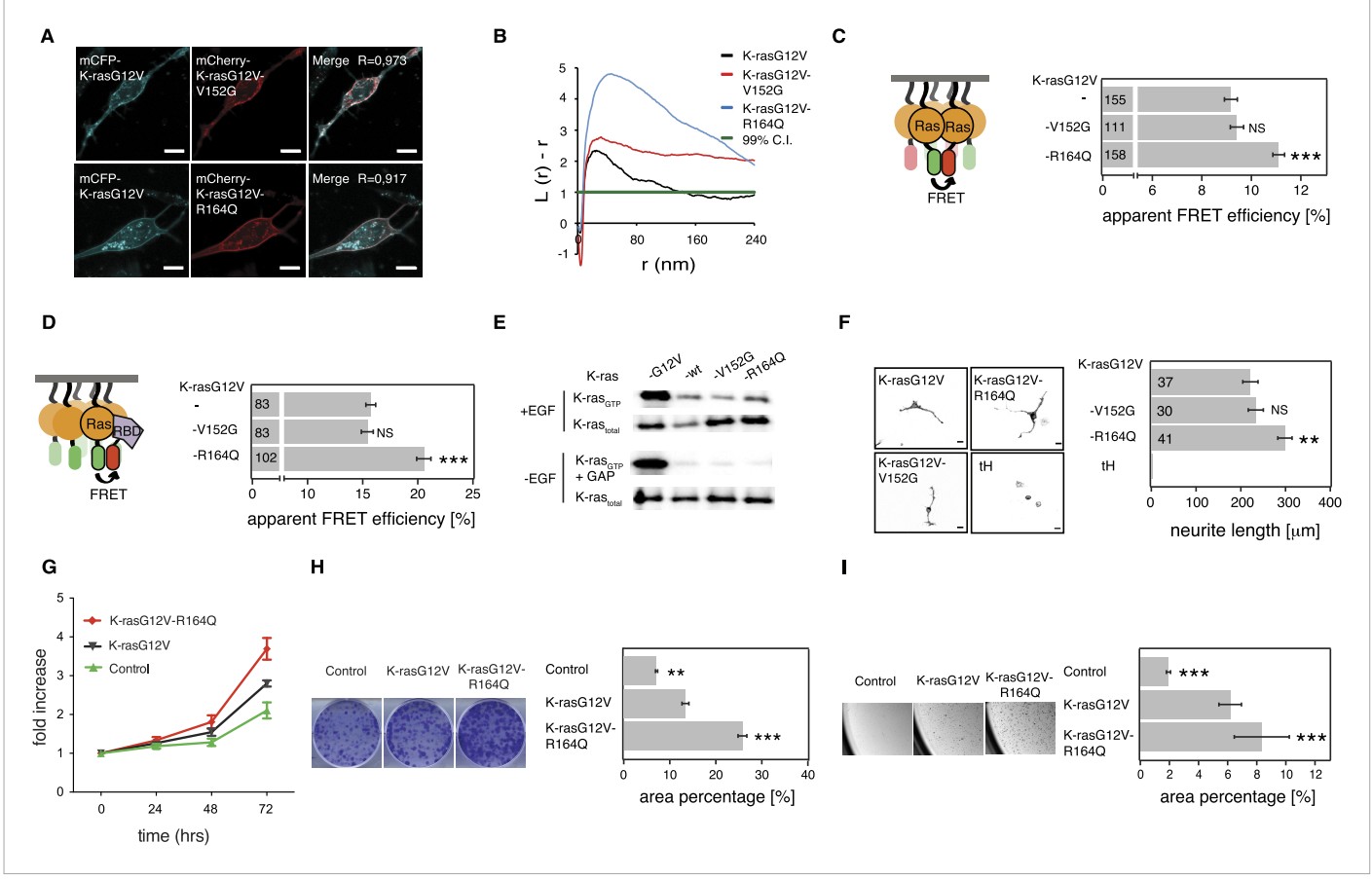

**Figure 6**. K-ras with colorectal cancer-associated mutation R164Q displays increased nanoclustering to drive oncogenic transformation. (**A**) Colocalization of mCFP-K-rasG12V and switch III mutants mCherry-K-rasG12V-V152G and mCherry-K-rasG12V-R164Q in BHK cells. The Manders' coefficient (*R*) that quantifies colocalization is marked on the merged images. Scale bar is 10 µm. (**B**) Electron microscopic nanoclustering analysis of BHK cells expressing indicated mGFP-tagged K-ras mutants. Normalized univariate K-functions, where maximal *L(r)-r* values above the 99% CI for complete spatial randomness indicate clustering at that value of r (number of membrane sheets analyzed per condition, n > 15). (**C**) The nanoclustering-FRET response of K-rasG12V-V152G, K-rasG12V-R164Q as compared to their parent construct in BHK cells. (**D**) RBD-recruitment FRET analysis of K-rasG12V-V152G, K-rasG12V-R164Q as compared to their parent construct in BHK cells. (**E**) RBD-pulldown experiments in BHK cells transiently expressing indicated K-ras mutants or (wild-type) wt K-ras. Top panel shows level of active K-ras after EGF-stimulation (100 ng/ml). Bottom panel shows GAP-sensitivity of wt and mutant K-ras proteins. (**F**) MAPK-dependent PC12 differentiation assay. Cells transiently expressed indicated mGFP-tagged K-rasG12V with or without mutations V152G and R164Q, or the control tH for 48 hr. Subsequently neurite length was assessed (left). Examples of confocal images of PC-12 cells expressing indicated constructs used for quantification (right). Scale bar is 15 µm. (**C**, **D**, **F**) Numbers in bars give number of analyzed cells from three independent experiments. Error bars represent the standard error of the mean (±SEM). Statistical analysis was performed as described in the 'Materials and methods' (NS, non-significant; *p < 0.05; **p < 0.01; ***p < 0.001). (**G**) Proliferation of NIH/3T3 cells stably expressing K-rasG12V, K-rasG12V-R164Q, or a control construct for 72 hr. Control is transduced with the GFP reporter only. Error bars represent the standard error of the mean (±SEM). (**H**) Transformation assay of NIH/3T3 cells virally transduced to stably express-indicated constructs and a GFP reporter. Foci formed after 7 days of growth were stained with crystal violet and mean (±SEM) foci areas from three biological repeats were quantified (right). Representative images of crystal violet stained foci of transduced NIH/3T3 cells (left). (**I**) Anchorage-independent growth of NIH/3T3 cells stably expressing K-rasG12V as compared to K-rasG12V-R164Q or a vector-only control. Colonies were imaged after 14 days (left), and the colony area was determined (right). Three independent biological repeats were performed, and each repeat was done in triplicate. Error bars represent the standard error of the mean (±SEM). See also *Figure 6—figure supplement 1*.

The following figure supplement is available for figure 6:

**Figure supplement 1**. K-ras with mutation R164Q has unaltered Gal-1-complexation and biochemical properties.

As for the two N-ras nanocluster mutations, we observed increased Ras/MAPK activity of K-rasG12V-R164Q by significantly increased PC12 cell differentiation (*Figure 6F*) but not on the phospho-protein level (*Figure 6—figure supplement 1D*). In agreement with this hyperactivation phenotype, NIH/3T3 cells expressing the mutant (*Figure 6—figure supplement 1E*) proliferated faster (*Figure 6G*), were

more transformed (*Figure 6H*), and grew more anchorage independently in soft agar (*Figure 6I*) than the K-rasG12V parent. Thus mutation R164Q in K-ras increases cell proliferation and tumorigenicity by its increased nanoclustering.

## Discussion

Here, we have described an unprecedented mechanism for the activation of a signaling protein in cancer. We showed that rare cancer-associated mutations in the switch III regions of H-, N-, and K-ras can increase effector engagement, while not changing the affinity to the effector. Instead the increase of the nanoscale concentration of Ras in nanocluster hyperactivates Ras to promote tumorigenic growth (*Figure 7*).

Our data at the outset suggested a connection between the H-ras conformers and nanoclustering. How could the conformation couple to nanoclustering? To address this question, we will here try to synthesize the most recent work on the structure and conformation of Ras in particular in the lipid environment. We will furthermore relate this to the assembly of switch III-mutated Ras into dimers and oligomers in nanoscale clusters in the membrane and ultimately to effector activation.

Molecular dynamics simulations suggest that Ras isoforms change their conformation via different intramolecular routes (*Gorfe et al., 2008*). An important shared feature is that the identical effector lobe of Ras, which contains the switch I, II, and III regions actuates the C-terminal helix α5. This might provide the mechanistic explanation of how the nucleotide exchange-induced structural change is converted into different Ras isoform-specific conformational equilibria on the membrane (*Abankwa et al., 2010*).

A milestone work by Mazhab-Jafari et al. has recently provided the first structural data on membrane bound K-ras4B (*Mazhab-Jafari et al., 2015*). Their data suggest at least two conformational states, with the GTP-state displaying an occluded effector-binding site. Conversely, in the GDP-state the G-domain contacts the membrane via helix α4 and α5 thus exposing the effector interaction site.

| | Mutants | Conformation | GAP | RBD + soluble Ras | Localization | Nanoclustering-EM | Nanoclustering-FRET | RBD + membrane Ras | PC12 | Proliferation | Transformation | Soft agar |
|---|---|---|---|---|---|---|---|---|---|---|---|---|
| H-ras | D47A,E49A | RED | ● | ● | ● | ↑ | ↑ 20% | ↑ 30% | N.D. | N.D. | N.D. | N.D. |
| | G48R | RED | ● | ● | ● | ↑ | ↑ 40% | ↑ 20% | N.D. | N.D. | N.D. | N.D. |
| | G48R,D92N | RED | ● | ● | ● | ↑ | ↑ 110% | ↑ 40% | N.D. | N.D. | N.D. | N.D. |
| N-ras | T50I | N.D. | ● | ● (*) | ● | N.D. | ● | ↓ -20% | ● | N.D. | N.D. | N.D. |
| | E49K | N.D. | ● | ● (*) | ● | ↑ | ↑ 10% | ↑ 20% | ↑ 30% | N.D. | N.D. | N.D. |
| | C51Y | N.D. | ● | ● (*) | ● | ↑ | ↑ 20% | ↑ 20% | ↑ 50% | ↑ | ↑ 70% | ↑ 30% |
| K-ras | V152G | N.D. | ● | ● (*) | ● | ● | ● | ● | ● | N.D. | N.D. | N.D. |
| | R164Q | N.D. | ● | ● (*) | ● | ↑ | ↑ 20% | ↑ 30% | ↑ 30% | ↑ | ↑ 100% | ↑ 40% |

**Figure 7**. Summary table of Ras switch III mutant properties studied in this work. The table summarizes major experimental results obtained in different assays that were used to characterize switch III mutants of H-ras (highlighted in green), N-ras (violet), and K-ras4B (blue). Note that in addition extensive biochemical characterization data of the H-ras mutants can be found in *Supplementary file 1*. Black circle dot indicates that no significant change was observed. Black arrows represent significant increase (up) or decrease (down) of quantified parameters as compared to the parent RasG12V control. The percentage of these changes is given in addition. The following columns (italics) report on: *Conformation*—predicted H-ras mutant conformation, according to *Figure 1A*; *GAP*—sensitivity of mutants to GAP-stimulated GTP hydrolysis; *RBD + soluble Ras*—binding of the C-Raf-RBD to Ras mutants in solution, which was either measured by fluorescence anisotropy with purified proteins or in BHK cells treated with compactin using FRET (annotated with *); *Localization*—change in colocalization of mutants relative to their parent constructs in BHK cells determined by confocal microscopy; *Nanoclustering-EM*—changes in Ras nanoclustering obtained by electron microscopy or -FRET in BHK cells; *RBD + membrane Ras*—binding of the C-Raf-RBD to Ras mutants in BHK cells measured by FRET; *PC12*—results of PC12-cell neurite outgrowth assay; last three columns report on Ras-mutant transformed NIH/3T3 cell *proliferation*, focus formation (*transformation*) and anchorage-independent growth (*soft agar*).

Also for N-ras low-resolution spectroscopic data support a nucleotide-dependent conformational change on the membrane (*Kapoor et al., 2012a*, *2012b*). Importantly, the K-ras conformational reorientation resembles the one that is here assumed for H-ras. Interestingly, an inverse role of helix α4 in stabilizing GDP- and GTP-conformations of H- and K-ras is implied, which matches to experimental data that we have provided earlier (*Abankwa et al., 2010*). Therefore H- and K-ras reorientation may work just in the opposite fashion.

However, the effect of the conformational change can be several fold. On the one hand, Ras dimer formation and oligomerization into nanocluster might become altered. Evidence for dimers of H-ras and N-ras in the membrane (*Lin et al., 2014; Güldenhaupt et al., 2012*) and K-ras even in solution (at micromolar concentrations) has recently been provided (*Muratcioglu et al., 2015*; *Nan et al., 2015*). The structural data of K-ras dimers suggest that K-ras might preferentially dimerize via specific surfaces in both the GTP- and GDP-state (*Muratcioglu et al., 2015*). Interestingly, the more stable β-dimer, which interfaces basically via the effector lobe of Ras, could prevent effector access. Similarly, coarse grain simulations of the clustering behavior of membrane-bound H-ras in two different conformations (basically corresponding to those in *Figure 1—figure supplement 1*) revealed that depending on the conformational state, preferred interfaces of the Ras proteins could be utilized (*Li and Gorfe, 2013*).

On the other hand, a mutation-induced conformational change may directly affect effector engagement, as has been proposed for H-ras (*Abankwa et al., 2008a*) and shown for K-ras on nanodiscs (*Mazhab-Jafari et al., 2015*). However, the mechanistic basis of how Ras activation is propagated is further complicated by the fact that we still lack detailed structural insight into for example, the precise activation of effectors, such as Raf. Ras-GTP nanoclusters are immobile, suggesting that a transient signaling machinery is built up, which is also coupled or trapped by the cytoskeleton (*Kusumi et al., 2012*). The (isoform- or mutation-) specific GTP-Ras conformational states on the membrane may differentially engage Raf-isoforms thus allosterically enabling specific Ras-Raf-isoform coupling (*Abankwa et al., 2010*). Here, the Ras-isoform-specific (acidic) membrane environment needs to be considered, as it is significant to retain clustered Raf (*Plowman et al., 2008*; *Cho et al., 2012*). Nanocluster formation can in addition be facilitated by nanocluster scaffolds, such as Gal-1. Gal-1 is typically dimeric suggesting that in nanoclusters, dimeric Ras–Raf units may exist. Of note, the lipid anchors of Ras alone are sufficient to drive higher order clustering of both Ras and Raf (*Nan et al., 2013*).

Additional support for a role of stoichiometric Ras–Raf complexes in nanocluster is lent by the fact that Raf dimerization inducing compounds can increase Ras nanoclustering (*Cho et al., 2012*). Therefore, dimeric signaling units may be both the outcome, as well as the driver of multimeric nanocluster formation. This nanoclustering promoting activity of Raf-inhibitors might underlie the paradoxical hyperactivation of Ras signaling in Ras mutant tumors. Intriguingly, these Raf inhibitors act Ras isoform specifically on K- and N-ras-nanoclustering, suggesting that additional factors break the promiscuity of 'dimer inducers'. Given that membrane anchored (nanoclustered) Raf is constitutively active (*Nan et al., 2013*), facilitation of this effector clustering process is critical for downstream signaling output.

In the light of the complexity of the events that lead from Ras activation to conformational changes, oligomerization and effector recruitment, we here refrained from making any conformational predictions for the cancer-associated mutations of N- and K-ras. These would have to be experimentally established by high-resolution methods. We therefore limited ourselves to nanoclustering as an activity predicting observable.

If Ras dimerization or multimerization into nanocluster is so crucial for signal propagation, why is it not more frequently exploited in cancer? First of all, next to mutations in Ras that we have just described, one would have to take mutations and expression changes in largely unknown nanocluster scaffolds into account. Secondly, based on our data Ras activity is typically up-tuned 2–3 fold by nanoclustering (*Guzmán et al., 2014b*). By contrast, GEF activation switches Ras on in an 'all-or-nothing' fashion (and GAPs off in the same manner), changing the enzymatic activity of Ras by 100 or even >50,000-fold (*Eccleston et al., 1993*; *Ford et al., 2006*). This would approximate the relative difference in Ras activity induced by hot-spot mutations and is much higher than activity changes by switch III-nanocluster mutations. Correspondingly, one could expect that this relation be to some extent reflected in the relative abundance (~1/2300) of switch III mutations in cancer. Currently it is unclear, whether there is a particular relevance for switch III mutations in certain cancer types, as the frequencies are too low for any conclusive statement. Likewise, it is unclear whether these mutations represent remnants of early mutational events in tumors or stem from certain niches of it. Given the large activity difference, it is

obvious that switch III mutated clones would easily be outcompeted by Ras hot-spot-mutated clones in a tumor. These features combined with the fact that switch III mutations may be associated with conformational changes in Ras that alter effector engagement, suggest that these mutations belong to the recently proposed class of 'latent driver' mutations (*Nussinov and Tsai, 2015*). Latent drivers are non-hotspot, lower frequency mutations, which appear like passenger mutations in that they do not cause malignancy. However, it is speculated that in a certain context they could provide a growth advantage.

In addition to cancer samples with switch III mutations, we also investigated two mutations that are found in RASopathies. It was previously speculated that the Noonan syndrome-associated mutation N-ras-T50I might be linked to the orientation-switch III mechanism (*Cirstea et al., 2010*). However, our data on this mutation do not support a gain-of-function downstream of GTP-loaded Ras (*Figure 5C–F*), neither could we observe any change in EGF-dependent Ras activation (*Figure 5E*). Likewise, we did not observe any changes by mutation V152G in K-ras (*Figure 6*). Thus RASopathy-associated switch III mutations do not show changes in nanoclustering, but may have unknown alterations (N-rasT50I, *Figure 5* and [*Cirstea et al., 2010*]; K-rasV152G, *Figure 6*; K-ras-E153V and K-ras-F156L [*Gremer et al., 2011*; *Mazhab-Jafari et al., 2015*]).

Given the evidence for nanoclustering of a number of other signaling proteins, such as GPI-anchored proteins (*Goswami et al., 2008*), heterotrimeric G proteins (*Abankwa and Vogel, 2007*), Rho- and Rab-small GTPases (*Köhnke et al., 2012*), as well as Src-family kinases (*Najumudeen et al., 2013*), it is timely to consider this mode of activity regulation as a new fundamental building block in the cellular signaling architecture.

## Materials and methods

### DNA constructs and molecular cloning

Plasmids pmGFP-H-rasG12V, pmGFP-H-RasG12V-D47A,E49A, pmGFP-H-rasG12V-R169A,K170A, pmGFP-H-rasG12V-R128A,R135A, pmCFP-K-rasG12V, pmCit-N-rasG12V, pmRFP-C-Raf-RBD, and mRFP-Gal-1 were described in *Abankwa et al. (2008)*, (*2010*). Plasmid pcDNA3-asGal-1 is an antisense construct and was used to deplete Gal-1, while plasmid pcDNA3-Gal-1 was used to overexpress Gal-1 as described in *Paz et al. (2001)*. Plasmid pmGFP-tH was described elsewhere (*Zhou et al., 2012*).

Site-directed mutagenesis was done by GenScript USA Inc. (Piscataway, NJ, USA) to introduce the specific mutations. To generate plasmids pmGFP-H-rasG12V- R128A,R135A,D47A,E49A mutations, R128A and R135A were introduced to the plasmid pmGFP-H-rasG12V-D47A,E49A and for pmGFP-H-ras-R161A,R169A,K170A, mutation R161A to the plasmid pmGFP-H-ras-R169A,K170A. For pmGFP-H-ras(wt) plasmid pmGFP-H-rasG12V was backmutated to G12G.

Plasmids pQE-A1 and pQE-A1-H-ras(wt) were described in *Guzmán et al. (2014b)*. To generate plasmids pmGFP-H-ras-G48R, pmGFP-H-rasG12V-G48R, pQE-A1-H-ras-G48R,D92N, pmGFP-H-rasG12V-G48R,D92N, mutations G48R alone or together with D92N were introduced to the plasmids pmGFP-H-rasG12V, pmGFP-H-ras(wt), and pQE-A1-H-ras(wt), respectively. H-ras-D47A,E49A and H-ras-G48R from pmGFP-H-ras-D47A,E49A and pmGFP-H-rasG48R were subcloned into the KpnI and BglII restriction sites of the pQE-A1 plasmid.

Plasmids pGEX-4T1-huNF1-333 encoding the catalytic domain of human neurofibromin GAP, NF1-333 (comprising residues 1198–1530), and pGEX-2T-huC-RAF-RBD for expression of GST-C-Raf-RBD were described previously (*Herrmann et al., 1995*; *Gremer et al., 2011*).

Plasmid pmCit-N-rasG12V was first backmutated to generate pmCit-N-ras(wt) and both plasmids then served as a template to which mutations T50I, E49K, and C51Y were introduced in order to generate pmCit-N-ras-T50I, pmCit-N-rasG12V-T50I, pmCit-N-ras-E49K, pmCit-N-rasG12V-E49K, pmCit-N-ras-C51Y, and pmCit-N-rasG12V-C51Y. Plasmids pmCFP-K-ras-R164Q, pmCFP-K-rasG12V-R164Q, pmCFP-K-ras-V152G, and pmCFP-K-rasG12V-V152G were constructed by site-directed mutagenesis on the plasmids pmCFP-K-rasG12V and pmCFP-K-ras(wt), which was first generated by backmutating the pmCFP-K-rasG12V (by GenScript Inc.). To generate pmGFP-K-rasG12V, pmGFP-K-rasG12V-R164Q, pmGFP-K-rasG12V-V152G, pmGFP-N-rasG12V, pmGFP-N-rasG12V-T50I, pmGFP-N-rasG12V-E49K, and pmGFP-N-rasG12V-C51Y, mGFP from pmGFP-H-rasG12V was subcloned into the NheI and BsrGI restriction sites of the corresponding pmCit-N-ras and pmCFP-K-ras constructs.

To generate pmCherry-H-rasG12V, pmCherry-H-rasG12V-D47A,E49A, pmCherry-H-rasG12V-G48R, pmCherry-H-rasG12V-G48R,D92N, pmCherry-K-rasG12V, pmCherry-K-rasG12V-R164Q, pmCherry-K-rasG12V-V152G, pmCherry-N-rasG12V, pmCherry-N-rasG12V-T50I, pmCherry-N-rasG12V-E49K, and

pmCherry-N-rasG12V-C51Y, mCherry from pmCherry-C1 plasmid (Clontech Laboratories Inc., Mountain View, CA, USA) was subcloned into the NheI and BsrGI restriction sites of the corresponding pmGFP-H-ras, pmCit-N-ras, and pmCFP-K-ras constructs. To generate pmCFP-H-rasG12V and pmCFP-N-rasG12V, mCFP from pmCFP-K-rasG12V was subcloned into the NheI and BsrGI restriction sites of the corresponding pmGFP-H-rasG12V and pmCit-N-rasG12V.

All fluorescent protein-tagged Ras constructs also contained a HA-tag, which is located between the sequence of the fluorescent tag and that of Ras. All constructs were verified by sequencing (GATC Biotech, Cologne, Germany and GenScript USA Inc.).

## Cell culture

Baby Hamster Kidney (BHK) and NIH/3T3 cells were grown in Dulbecco's modified Eagle medium (DMEM) supplemented with 10% FBS (not heat inactivated in case of NIH/3T3 cells), L-glutamine, penicillin (100 U/ml), and streptomycin (100 µg/ml). PC-12 cells were grown on plates coated with 50 µg/ml of rat tail collagen I (Termo Fisher Scientific Gibco, Waltham, MA, USA) in Roswell Park Memorial Institute (RPMI) 1640 medium supplemented with 10% horse serum, 5% FBS, L-glutamine, penicillin (100 U/ml), and streptomycin (100 µg/ml). All cell types were typically grown to a confluency of 80% ($8 \times 10^7$ cells/ml) and then sub-cultured every 2–3 days. BHK and PC-12 cell lines were obtained from ATCC cell repository (LGC Standards GmbH, Wesel, Germany). Cells were cultured for not more than 10 passages. NIH/3T3 cell line was obtained from cell line collection of VTT Technical Research Centre of Finland.

## Electron microscopic analysis of Ras nanoclustering

Immuno electron microscopy spatial mapping was performed as described previously (*Prior et al., 2003a*, *2003b*). Apical plasma membrane sheets were prepared from BHK cells transiently expressing mGFP-tagged Ras mutants, fixed with 4% PFA, 0.1% glutaraldehyde and labeled with 4.5 nm (diameter) gold nano-particles coupled to anti-GFP antibody. Digital images of the immunogold-labeled plasma membrane sheets were taken at 100,000× magnification in an electron microscope Jeol JEM-1400 (JEOL USA, Inc., Peabody, MA, USA). Intact 1 µm² areas of the plasma membrane sheet were identified using ImageJ and the (x, y) coordinates of the gold particles were determined as described. A minimum of 17 plasma membrane sheets were imaged and analyzed for each Ras mutant. A bootstrap test constructed as previously described is then used to evaluate the statistical significance of differences between replicated point patterns (*Plowman et al., 2005*).

## FRET-imaging using fluorescence lifetime microscopy

BHK cells were transfected using Fugene 6 transfection reagent (Promega Biotech AB, Madison, WI, USA) with the donor alone (mGFP-tagged Ras constructs) in control samples, or mGFP-tagged Ras together with the acceptor. Acceptors were mCherry-tagged Ras constructs in nanoclustering-FRET experiments, mRFP-C-Raf-RBD in RBD-recruitment FRET experiments or mRFP-Gal-1 in cellular Gal-1 complexation experiments. To modulate cellular Gal-1 levels co-transfections with pcDNA3-asGal-1 or pcDNA3-Gal-1 were done, vectors which express antisense and sense Gal-1 sequences to deplete and increase Gal-1 levels in cells, respectively. Treatment with 5 µM compactin was done 4 hr after transfection. After 48 hr, cells were fixed with 4% PFA and mounted with Mowiol 4–88 (Sigma–Aldrich, St. Louis, MO, USA). The mGFP fluorescence lifetime was measured using a fluorescence lifetime imaging attachment (Lambert Instruments, Groningen, Netherlands) on an inverted microscope (Zeiss AXIO Observer D1, Jena, Germany). For the sample excitation sinusoidally modulated 3 W, 497 nm LED at 40 MHz under epi-illumination was used. Cells were imaged using the 63×, NA 1.4 oil objective with the GFP filter set (excitation: BP 470/40, beam splitter: FT 495, emission: BP 525/50). The phase and modulation were determined using the manufacturer's software from images acquired at 12 phase settings. Fluorescein at 0.01 mM, pH 9 was used as a lifetime reference standard. Per condition, the fluorescence lifetime was measured typically for >40 cells from three biological repeats. From the obtained fluorescence lifetimes the apparent FRET efficiency was calculated as described in *Guzmán et al. (2014b)*.

## Bioinformatics methods

For a comprehensive analysis of cancer-associated mutations in the orientation-switch III region, point mutation data for Ras isoforms detected either in clinical tumors or cancer cell lines were downloaded from public databases: cBioPortal to access data from TCGA and other cancer genomic studies

(http://www.cbioportal.org/), COSMIC v68 (http://cancer.sanger.ac.uk/) and ICGC data portal (https://dcc.icgc.org/). Frame-shifts, insertions, deletions, and stop codon mutations in the Ras isoforms were removed from the analyses. Point mutations, both synonymous and non-synonymous, were categorized into the 4 regions based on the following criteria: β2-β3-loop (residues of Ras 40–56), helix α4 (122–138), helix α5 (152–166), hvr (166–188/9). For those samples that were redundant across the databases, the union of the observed point mutations was reported for the co-mutated genes.

For analysis of phylogenetic conservation of computationally and experimentally suggested switch III residues D47 and E49, the NCBI's Batch-Conserved Domain interface (http://www.ncbi.nlm.nih.gov/Structure/bwrpsb/bwrpsb.cgi) was used to identify proteins similar to that of human H-, N-, and K-ras isoforms. The top 17 hits were further analyzed in the MacVector software (MacVector Inc., North Carolina, USA). Sequences were first aligned using the ClustalW algorithm and the phylogenetic trees of the switch III elements β2-β3-loop (residues of human Ras 40–56) and helix α5 (residues 152–166) were reconstructed using unrooted neighbor-joining method. The bootstrap probability was calculated from consensus of 1000 bootstrap samples. The co-evolutionary distances of β2-β3-loop and helix α5 regions were compared using the *Mirror tree* web server (http://csbg.cnb.csic.es/mtserver/) and displayed as the correlation coefficient with assigned statistical significance p values.

## Western blot analysis of MAPK-pathway activity

To analyze the activity of mutant H-rasG12V-D47A,E49A, BHK cells were transiently transfected with vector control (pC1 vector, Promega Biotech AB), mGPF-H-rasG12V or mGFP-H-rasG12V-D47A,E49A using Lipofectamine (Thermo Fischer Scientific Invitrogen). After 16 hr, cells were serum starved for 2 hr. Whole cell lysates were separated by SDS-PAGE and blotted using Cell Signalling Technology (Boston, MA, USA) antibodies against pC-Raf (Ser338) (9427S), pMEK (9154S), pERK (4695S), total C-raf, GFP and actin. Band intensities were imaged using FluorChem Q System (ProteinSimple, San Jose, CA, USA), quantified by densitometry and normalized to the average intensity values of all measured conditions. Averages were calculated from three independent biological repeats.

To analyze the activity of the K-ras and N-ras switch III mutants, BHK cells were transiently transfected with vector control (pEGFP-N1 vector, Clontech laboratories Inc.), pmCFP-K-ras(wt), pmCFP-K-ras-V152G, pmCFP-K-ras-R164Q, pmCit-N-ras(wt) pmCit-N-ras-E49K, pmCit-N-rasT50I, and pmCit-N-rasC51Y using JetPRIME transfection reagent (Polyplus-transfection, New York, NY, USA). After 24 hr, cells were serum starved for 5 hr and stimulated with 100 ng/ml EGF for 10 min. Whole cell lysates were first separated using SDS-PAGE and blotted using primary antibodies for MEK1/2 (9126), pMEK1/2 (9121), ppERK1/2 (9101), ERK1/2 (9102), AKT (9272, Cell Signalling Technology), pAKT1 (MAB7419, R&D Systems, Minneapolis, MN, USA), GFP (3999-100, BioVision, Inc., Milpitas, CA, USA), and β-actin (A1978, Sigma–Aldrich). Membranes were then probed with secondary peroxidase-conjugated IgG antibodies (sc-2370, sc-2954, Santa Cruz Biotechnology, Inc., Dallas, TX, USA). Chemiluminescence was detected using ChemiDoc MP System (Bio-Rad, Hercules, CA, USA).

## Protein production for biochemical studies

Wild-type and mutant H-ras as well as C-Raf-RBD proteins were prepared from *Escherichia coli* using the pQE-expression system (Qiagen, Hilden, Germany) as described (*Guzmán et al., 2014b*). The C-Raf-RBD and GAP-domain of neurofibromin comprising residues 1198–1530 of human NF1 (NF1–333) were produced as glutathione S-transferase (GST) fusion proteins in BL21 strain of *E. coli* from plasmids pGEX-4T1-Ntev-NF1-333 and pGEX-2T-C-Raf-RBD. All proteins were purified as described previously (*Guzmán et al., 2014b*).

## In vitro mant-GTPγS/Ras binding by fluorescence anisotropy measurements

Fluorescence anisotropy was used to study the affinity between mant-GTPγS and H-ras mutants. Measurements were conducted at room temperature in a buffer containing 25 mM Hepes pH 7.2, 100 mM NaCl, and 5 mM EDTA. The reaction was monitored by the anisotropy of 100 nM mant-GTPγS (BP 340/30 nm excitation filter and BP 485/20 nm emission filter) in the presence of H-ras mutants (from 0 nM to 3000 nM) using a Synergy H1 hybrid fluorescence plate-reader (BioTek, Winooski, VT, USA) equipped with a polarization filter cube. Data processing was done as described using the concentration of H-ras as X and the concentration of mant-nucleotide as $L_t$ (*Guzmán et al., 2014b*).

## In vitro RBD/Ras binding by fluorescence anisotropy measurements

Fluorescence anisotropy was used to study the affinity between the Ras binding domain of C-Raf (RBD) and H-ras mutants in solution. The experimental part and the data processing were done as described in *Guzmán et al. (2014b)*.

## SOS-dependent nucleotide association and dissociation

The GEF (histidine tagged SOS$^{cat}$, amino acids 564–1049) mediated Ras nucleotide exchange kinetics and data analysis were performed as described (*Kopra et al., 2014*). In brief, the homogeneous $Eu^{3+}$-GTP exchange assays were performed using the quenching resonance energy transfer (QRET) technique. In association experiments, H-Ras (200 nM), $Eu^{3+}$-GTP (10 nM), Quench II (22 µM) (QRET Technologies, Turku, Finland) were incubated for 5 min before $Eu^{3+}$-GTP association was triggered by SOS (1:1 H-Ras:SOS ratio) addition. In the $Eu^{3+}$-GTP dissociation assay, the $Eu^{3+}$-GTP association to H-Ras (200 nM) was first performed in the presence of SOS (200 nM). After 20 min of incubation the $Eu^{3+}$-GTP dissociation was induced with 100 µM GTP. All the measurements were performed using a Victor2 1420 multilabel counter (Perkin Elmer Life and Analytical Sciences, Turku, Finland) to record time-resolved luminescence data (emission 615 nm, excitation 340 nm, delay 400 µs and window time 400 µs). The kinetic data were analyzed using GraphPad Prism 6 software from GraphPad Software Inc. (La Jolla, CA, USA).

## RBD-pulldown experiments and GAP assay

GST-RBD pulldown experiments were performed essentially as described (*Gremer et al., 2011*). In brief, BHK cells were transiently transfected with the respective plasmids (mGFP-tagged H-ras or mCFP-tagged K-ras or mCit-tagged N-ras constructs) using JetPRIME transfection reagent (Polyplus-transfection, New York, NY, USA). 24 hr after transfection cells were serum starved for 4 hr and half of them stimulated with 100 ng/ml EGF for 5 min. Ras-GTP levels were determined using GST-RBD, the GST-fusion of the Ras binding domain of C-Raf, to pulldown active GTP-bound Ras from cellular lysates by glutathione beads (Pierce, Thermo Fisher Scientific, Waltham, MA, USA). Cells were lysed in lysis buffer (Tris/HCl pH 7.5, 100 mM NaCl, 2 mM $MgCl_2$, 1% Nonidet-40, 10% glycerol, EDTA-free inhibitor cocktail). In order to assess GAP-sensitivity of Ras mutants, the cell lysates were incubated in the presence and in the absence of 10 µg purified catalytic domain of NF1 (NF1-333) for 30 min at +4°C and then mixed with glutathione beads coupled to GST-RBD. The samples were washed four times with lysis buffer and subjected to SDS-PAGE (15% polyacrylamide). Bound Ras proteins were detected by western blotting using monoclonal antibodies against the HA-tag present in Ras constructs (HAtag antibody, C29F4, Cell Signaling Technology).

## Colocalization analysis of confocal images

To compare the subcellular distribution of Ras-proteins in BHK cells, co-expressing mCherry-tagged mutants with their mCFP-tagged parent Ras constructs confocal imaging using Zeiss LSM 780 was done (63×, NA 1.2 water immersion objective, mCFP excitation at 405 nm and mCherry excitation at 543 nm). To determine co-localization, the overlap Manders' coefficient ($R$) was calculated using the 'Manders' coefficient' ImageJ plugin developed by Tony Collins, Wayne Rasband and Kevin Baler (Wright Cell Imaging Facility, Toronto, Canada).

## PC-12 cell differentiation assay

PC-12 differentiation is a sensitive measure of MAPK-activity (*Cowley et al., 1994*; *Vaudry et al., 2002*). PC12 cells were seeded at a density of $10^4$ cells per well of a 4 well Lab-Tek II Chambered Coverglass (Thermo Fischer Scientific Nunc) coated with 0.1% of rat tail collagen I (Termo Fischer Scientific Gibco) in 30% ethanol. After 24 hr, cells were transfected with GFP-tagged Ras constructs or control construct mGFP-tH (tH representing the minimal membrane anchor of H-ras [*Plowman et al., 2005*]), using JetPRIME transfection reagent (Polyplus-transfection). GFP-positive cells that exhibited neurites with a length of at least two times the diameter of the cell body were imaged using the confocal microscope Zeiss LSM 510 META, 40×, NA 1.4 oil immersion objective, excited at 488 nm. Neurite length was determined using the ImageJ plugin 'NeuronJ' (*Meijering et al., 2004*). For each experimental condition at least 25 cells from three biological repeats were analyzed.

## Generation of NIH/3T3 cells stably expressing Ras mutants

To generate pooled clones of NIH/3T3 cells stably expressing N-rasG12V, N-rasG12V-C51Y, K-rasG12V, and K-rasG12V-R164Q, cells were transduced with lentiviral particles for expression of HA-tagged Ras-proteins under CMV promoter and GFP-Puromycin marker under RSV promoter (AMS Biotechnology, Abingdon, UK). As a control, cells stably expressing only GFP-Puromycin marker were used. Selection with puromycin (1 µg/ml) started 72 hr after infection. After selection, cells were expanded and used in assays as indicated. Western blotting with anti-HA-tag antibody (Cell Signalling, C29F4) was performed to verify stable and comparable protein expression levels of Ras mutants.

## Cell proliferation, focus formation, and anchorage-independent growth assays

In cell proliferation, dose response, focus formation and anchorage-independent growth assays NIH/3T3 cells stably expressing indicated Ras constructs were used. In proliferation assays, 500 cells were seeded per well in a 96-well plate and grown for 72 hr. At intervals of 24 hr, 15 µl AlamarBlue (Thermo Fisher Scientific Invitrogen) was added to each well, and after 3 hr of incubation, fluorescence intensity was measured at an excitation of 570 nm and an emission of 590 nm using a Synergy H1 Hybrid Reader (BioTek). The fluorescence values at each timepoint were normalized to the value at 0 hr and plotted. The error bars represent ±SEM. For each experimental condition, data are averages of data points that were acquired in hexaduplicate in each of three biological repeats. In a focus formation assay, 1,500 cells were seeded per well of a 6-well plate. After 7 days of growth, cells were fixed with 4% PFA, stained with 0.5% crystal violet in 10% ethanol for 15 min and washed with PBS to remove excess stain. The average colony area percentage from three independent biological repeats was calculated using the 'ColonyArea' ImageJ plugin (Guzmán et al., 2014a). In the anchorage-independent colony formation assay, which correlate typically with in vivo tumorigenicity (Shin et al., 1975). 50,000 cells were resuspended in growth medium containing 0.4% agarose (4% Agarose Gel, Termo Fisher Scientific Gibco; top layer) and plated on a bottom layer containing growth medium and 1.2% agarose in a 6-well plate. After 10–14 days of growth, cells were fixed in methanol/acetone (1:1) and washed with PBS. Colonies were imaged using a Zeiss SteREO Lumar V12 stereomicroscope. Analysis was done using the ImageJ software. First, the background was subtracted using the rolling ball function with a radius of 50 µm, then auto-tresholding was applied to separate the colonies. Area percentage was calculated using the ImageJ built-in function 'Analyze Particles' with exclusion of particles smaller than 500 µm$^2$ that are not considered colonies. Statistical differences were determined by two-way ANOVA test.

## Statistical analysis

For all experimental data, statistical differences were determined using an analysis of variance (ANOVA) complemented by Tukey's honest significant difference test (Tukey's HSD). The software R version 2.15.2 (R Development Core Team, Vienna, Austria) was used to perform these analyses. Statistical significance levels are annotated as NS = non significant, that is, $p > 0.05$, *$p < 0.05$, **$p < 0.01$, ***$p < 0.001$.

## Acknowledgements

We thank Yonatan Gebremariam Mideksa for sequence alignments and phylogenetic analysis, Jasmin Varjonen for experimental support, Dr Christina Oetken-Lindholm and Dr Elina Siljamäki for help with virus work. We thank Dr Christian Rupp for help with the soft-agar assay. Plasmids encoding the catalytic domain of human neurofibromin 1 GAP (NF1-333) and GST-C-Raf-RBD were kindly provided by Dr Mohammad Reza Ahmadian. This work was supported by the Academy of Finland fellowship grant, the Sigrid Juselius Foundation, the Cancer Society of Finland, the Marie-Curie Reintegration Grant and the Jane and Aatos Erkko Foundation grants to DA. MS was supported by the TUBS graduate school. Texas Digestive Diseases Center Pilot and Feasibility Grant (P30 DK56338) to YZ and CPRIT (Cancer Prevention Research Institute of Texas) grant RP130059 to JFH. AJ Integrative Life Science (ILS) doctoral program, TA Academy of Finland, Cancer Society of Finland.

## Additional information

### Funding

| Funder | Grant reference | Author |
|---|---|---|
| Suomen Akatemia (Academy of Finland) | 272437, 269862, 279163 | Tero Aittokallio |
| Sigrid Juséliuksen Säätiö | N.A. | Olga Blazevits, Benoit Lectez |
| Syöpäjärjestöt | N.A. | Alessio Ligabue, Tero Aittokallio |
| Suomen Akatemia (Academy of Finland) | 252381, 256440, 281497,141516 | Alessio Ligabue, Camilo Guzman, Daniel Abankwa |
| Turku Doctoral Programme of Biomedical Sciences | Graduate Student Fellowship | Maja Solman |
| Texas Medical Center Digestive Disease Center | P30 DK56338 | Yong Zhou, Hong Liang |
| Cancer Prevention and Research Institute of Texas (CPRIT) | RP130059 | John F Hancock |
| Integrative Life Science Doctoral Program (ILS) | Graduate Student Fellowship | Alok Jaiswal |
| Suomen Akatemia (Academy of Finland) | 138584, 270010 | Harri Härmä |
| National Doctoral Programme in Informational and Structural Biology | N.A. | Kari Kopra |

The funders had no role in study design, data collection and interpretation, or the decision to submit the work for publication.

### Author contributions

MS, Acquired and analyzed FRET and cellular data, designed the experiments and wrote the manuscript; AL, Analyzed nucleotide exchange data, purified proteins, acquired and analyzed anisotropy binding data; OB, Acquired and analyzed GAP-dependent and independent Ras-activity, acquired anisotropy binding data and purified proteins; AJ, Carried out the bioinformatic analyses and cancer genome database mining; YZ, Acquired, analyzed and designed EM experiments; HL, Performed western blot experiments for MAPK signaling of H-ras mutants; BL, Performed western blot experiments for MAPK signaling of K-ras and N-ras mutants; KK, Acquired nucleotide exchange data; CG, Analyzed confocal microscopy and anchorage-independent growth data, and assisted in analysis of PC12 cell and focus formation data and statistics; HH, Conceived the nucleotide exchange assay; JFH, Designed EM experiments; TA, Contributed the bioinformatic analyses; DA, Conceived the project, designed the experiments and wrote the manuscript

## Additional files

### Supplementary files

• Supplementary file 1. Thermodynamic and kinetic parameters of H-ras mutant in vitro experiments. Table contains background corrected GEF-dependent $Eu^{3+}$-GTP association (kon), and dissociation (koff) kinetics of wt H-ras and mutants. The Kon, Koff, and Kd derived from Koff/Kon values are calculated from corrected association and dissociation data as described in 'Materials and methods'. Dissociation constants (Kd) between mantGTPγS and wt H-ras and its mutants (in the presence of EDTA and in the absence of $Mg^{2+}$). The absence of $Mg^{2+}$ increases the Kd, which is otherwise in the pM range (*John et al., 1988*, *1990*). Dissociation constants (Kd) of C-Raf-RBD and mantGTPγS-bound wt H-ras and its mutants were measured by fluorescence anisotropy as described in 'Materials and methods'. Raw data are shown in *Figure 2—figure supplement 2*.

• Supplementary file 2. Distribution of isoform specific coding mutations in the switch III region by cancer tissue type. Table details the frequency of amino acid changes due to coding mutations observed in switch III region of each isoform in different tissue types of cancer. Point mutations leading to different amino acid residue changes at the same coding position have been added to indicate the number of changes at that position. Summary for total mutations observed in all isoforms and total mutations per isoform have also been provided. Gray highlighted cells are the tissue types and switch III regions having the highest number of coding mutations at that position. Red and bold highlighted numbers indicate coding mutations observed in patient samples with the corresponding cancer tissue type.

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
