## [Decision Letter]

Thank you for submitting your work entitled “Cancer associated mutations in the switch III-region of Ras increase tumorigenicity by nanocluster augmentation” for peer review at *eLife*. Your submission has been favorably evaluated by Tony Hunter (Senior Editor) and three reviewers, one of whom is a member of our Board of Reviewing Editors.

The reviewers have discussed the reviews with one another and the Reviewing Editor has drafted this decision to help you prepare a revised submission.

The paper explores the roles of mutations in the newly identified “switch III” region of Ras. Such mutations are found at low frequency in human tumors. They were predicted to potentially affect Ras orientation in the membrane and nanoclustering. The authors find that the mutations studied do significantly affect nanoclustering and effector recruitment in cells, while having little effect on biochemical measures of Ras activity in vitro. The results support the concept that increased nanoclustering may be selected for in cancer.

The reviewers do not request major new experiments, however, it is clear from the reviews (provided below) that the presentation requires improvement. The reviewers have discussed their individual comments and arrived at the following consensus:

1) The initial models should be presented more clearly (ideally naming the two conformations (e.g. A and B, or red and blue as in Figure 1) and distinguishing the conformation from the degree of clustering), and the data should be summarized (e.g. in a table documenting conformation, clustering, RBD recruitment and other assays for each mutant) so as to help the reader follow the interpretation. The table should include the Noonan mutations for which you find no evidence for activation.

2) In light of recent developments, please revise your Discussion to put your observations into structural context.

3) Please reconsider the title. “Cancer associated mutations in the switch III region of Ras increase tumorigenicity by nanocluster augmentation” sounds inclusive, implying that all cancer-associated mutations in switch III have the same effect. Your results indicate a more nuanced conclusion.

Reviewer #1:

Ras family proteins are closely related in basic structure and all are transforming, yet they differ in their utilization of different effectors and their mutation spectra in cancer. Understanding the nuances of Ras activity is undoubtedly important scientifically and medically. This paper focuses on unusual Ras mutations found in rare cancers and RASopathies.

Abankwa and colleagues previously showed that Ras adopts alternative conformations at the plasma membrane, depending on its GTP state, stabilized by membrane interactions with basic residues in α4 and the hypervariable region (HVR). The alternative conformations have different access to effectors, hence affecting signaling even by GTPase-defective RasG12V. They previously proposed that guanine nucleotides regulate the orientation in the membrane through a conformation switch in β2-3 and α5 (“switch 3”), which is then relayed to α4 and HVR. Here they test that proposal by combining activating (D47A, E49A) and inactivating (R161A) switch 3 mutations with α4 and HVR mutations. Their results agree with their predictions. They also measured nanoclustering using EM and FRET approaches. They found that the activating mutations in switch 3 augment nanoclustering and RafRBD recruitment by RasG12V, and signaling is further stimulated by expressing galectin-1, which they showed before further augments nanoclustering. Increased RBD recruitment was not due to an increase in affinity of the purified Ras, but was only detected when membrane was present. However, it isn't clear how much of the increase in signaling is due to increased nanoclustering and how much is due to additional stabilization of the more accessible conformation.

The effects of Ras mutations on membrane orientation and effector binding were previously inferred by Abankwa and Hancock using indirect assays and molecular dynamics simulations. These inferences were recently borne out by the KRas4B.GDP and GMPPNP NMR structures that were recently published in PNAS, showing that the effector domain of Ras.GTP is partially occluded when bound to a model membrane (May 26 2015 issue, Mohammad T. Mazhab-Jafari, 6625-6630, doi:10.1073/pnas.1419895112). It isn't clear how these NMR structures compare with the models proposed by Abankwa and colleagues. The Mazhab-Jafari paper also tested the effects of RASopathy (Noonan) mutations 5N and D153V on effector binding, and found that the mutations slightly increased the affinity of membrane-associated Ras for effectors but had no effect on free Ras. They proposed that decreased autoinhibition may account for the increased activity of these mutants. Mazhab-Jafari et al. did not address nanoclustering and did not test the same RASopathy mutations as have now been tested in the paper under review. Nevertheless, the new Mazhab-Jafari paper will lessen the impact of the present paper. New concepts have not been revealed.

Reviewer #2:

This paper explores the role of mutations in the switch III region of all Ras isoforms that may contribute to control of conformation and nanoclustering. The scope is to examine mutations that are predicted by earlier modeling and that exist naturally in tumors using electron microscopy and FRET to assess clustering and effector recruitment. Some analysis of signaling and transformation is also performed. Reading the paper is difficult because there is an ambiguity associated with discussing the conformers and if these are independent of the GTP/GDP state they need a clear definition for each, like A and B, etc. They refer to “conformer” in several places without specifying which conformer they are referring to, or as if there is a non-conformer or conformer state. They general fix the major which in the on position using the G12V mutant and then determine if mutations modulate self-FRET or RBD interactions. To me, these tools allow evaluation of clustering and RBD binding, but don't say much about conformation, which seems like a model to explain the findings, but they can otherwise make no strong conclusions. Another general comment is that they start with a strong emphasis on modeling based predictions, but then analyze several natural mutations with no prediction or reference back to the modeling to evaluate the negative and positive results obtained. In fact, they fail to provide any further comment when natural mutations in these regions have no effects on clustering or RBD binding. While the point is clearly to support the conformational switch III determinant in control of activity through nanoclustering, the analysis of the naturally existing mutations is not really pulled together to support this model.

Major issues:

1) The use of hetero-FRET to evaluate clusters with 4-6 molecules per cluster is not very sensitive and the changes in FRET efficiency are small. Homo-FRET based on polarization anisotropy would provide greater sensitivity for analysis of small clusters because the statistics of pairing donors and acceptors will be better.

2) Neither the hetero-FRET methods provide any direct insight into the proposed conformational changes. Is it not possible to directly evaluate the location of a probe integral to or associate with Ras and a membrane bound accepted (like a fluorescent lipid) to more directly assess the relationship of Ras to the membrane?

3) In the model driven mutational analysis they test certain combinations of mutations and find that all results are as they expect, but they don't test the entire matrix of combinations to show that combinations that they would not predict to be neutralizing show some dominance of the activating or inactivating mutation. Thus, “these results therefore corroborate that the switch III region is allosterically coupled to the reorientation of H-ras on the membrane” is not a strong conclusion.

4) The story would have more cohesion if they could make predictions or perhaps just provide some structural insight into the results with the cancer-associated mutations.

Reviewer #3:

This is an important work from a group that pioneered the discovery of Ras nanoclusters. Here the authors take another step associating mutations in the switch III region with nanocluster augmentation and signaling, thus revealing a novel mechanism for oncogenic mutations. This mechanism is both logical and supported by the large body of experimental data that they obtained. As such, this manuscript merits publication.

It would have been nice if the authors were able to suggest a structural mechanism illustrating this significance, that is, how the mutations might lead to the consequences that they observe, although I realize that this is (very) difficult.

Second, as the authors indeed note, Ras nanoclustering for signal propagation (assuming the primary the Raf/MAPK pathway) is not reflected in the current oncogenic Ras mutations in cancer genome databases. Can the authors suggest why?

One possibility could relate to a newly proposed mechanism of allosteric ‘latent’ mutations (‘Latent drivers’ expand the cancer mutational landscape. Nussinov R, Tsai CJ. Curr Opin Struct Biol. 2015 doi: 10.1016/j.sbi.2015.01.004.). A ‘latent’ mutation description would fit quite well the moderate properties described here.

Third, I do not understand why in Figure 3 mutations are labeled with no changes for V45V, D173D, R164R, and L171L. Please, either explain or fix the figure.

The paper is comprehensive, well-written and illustrated and I think the proposition made is novel and makes sense.

---

## [Author Response]

*1) The initial models should be presented more clearly (ideally naming the two conformations (e.g. A and B, or red and blue as in*
Figure 1*) and distinguishing the conformation from the degree of clustering), and the data should be summarized (e.g. in a table documenting conformation, clustering, RBD recruitment and other assays for each mutant) so as to help the reader follow the interpretation. The table should include the Noonan mutations for which you find no evidence for activation*.

We thank the reviewers and editor for their constructive criticism. We have made several changes to the first part of the Results, modified the Figure 1 legend and added the requested table to clarify the raised issues.

a) We explain that we are mainly talking about conformers in the GTP-state and refer to them as ‘red and blue’ (in the subsection “The switch III region of H-ras couples to G-domain reorientation” and in the legend of Figure 1). There we also clarify that we do not exclude a direct effect of conformation on effector engagement. We have reorganized Figure 1, splitting the former panel (A) as a supplement to Figure 1 (Figure 1—figure supplement 1). We have also moved all tables with mutations to the top of the former (B, C), in line with the flow of the text in the manuscript and indicated where the model is based on existing or predicted data. We believe that these changes have now made this figure clearer.

b) We interpret our data with respect to nanoclustering and not conformation, for reasons that we extensively outline in the Discussion. Essentially, it is too complex to make inferences about the conformation, based on the nanoclustering data.

c) In addition, we have now prepared a summary table of all Ras mutants (Figure 7), which gives an overview of the qualitative results in the various assays that we performed.

*2) In light of recent developments, please revise your Discussion to put your observations into structural context*.

We have strived to discuss our work in the context of the most recent structural insights on Ras. In the Discussion, we synthesize recent structural, computational and microscopic data on Ras conformation, dimerization, nanoclustering and effector binding. We have included the suggested citations and added several other important citations.

More specifically, we discuss that conformation could affect Ras dimerization, and both could directly affect effector recruitment. We mention that effector activation in complex with Ras on the membrane is structurally unresolved, a major bottleneck in our understanding of Ras biology. We consider how nanocluster can be stabilized and that apparently effector dimers play a significant role.

We also discuss that the switch III mutations fit the criteria of ‘latent drivers’, which were proposed in the publication suggested by Reviewer 3 (please see in the Discussion: “If Ras di- or multimerisation […] could provide a growth advantage.”).

As Reviewer 3 pointed out, it is essentially not possible to predict conformational changes from our nanocluster data (see the Discussion: “In light of the complexity […] activity predicting observable”). Therefore, we speculate only for the H-ras switch III mutants in the new summary table on the conformation, as we have experimental evidence for this (Figure 1 and Figure 4—figure supplement 1).

Conscious of the general difficulty of connecting nanoclustering back to conformation, we focused our attention in the manuscript on Ras nanoclustering as the observable, which integrates any conformational or complexation (as additional binding partners may be recruited) change, while showing a tight correlation with functional output of Ras.

*3) Please reconsider the title. “Cancer associated mutations in the switch III region of Ras increase tumorigenicity by nanocluster augmentation” sounds inclusive, implying that all cancer-associated mutations in switch III have the same effect. Your results indicate a more nuanced conclusion*.

We agree and have accordingly changed the title to: “Specific cancer associated mutations in the switch III region of Ras increase tumorigenicity by nanocluster augmentation”.

We hope that this change is sufficient to reflect the fact that we have only studied a subset of (naturally occurring) mutations in that region.